# Effect of Paxlovid treatment during acute COVID-19 on Long COVID onset: An EHR-based target trial emulation from the N3C and RECOVER consortia

Alexander Preiss[1]*◦, Abhishek Bhatia[2]◦, Leyna V. Aragon[3], John M. Baratta[2], Monika Baskaran[2], Frank Blancero[4], Michael Daniel Brannock[1], Robert F. Chew[1], Iván Díaz[5], Megan Fitzgerald[6], Elizabeth P. Kelly[2], Andrea G. Zhou[7], Thomas W. Carton[8], Christopher G. Chute[9], Melissa Haendel[2], Richard Moffitt[10], Emily Pfaff[2], on behalf of the N3C Consortium and the RECOVER Cohort

1 RTI International, Durham, North Carolina, United States of America, 2 University of North Carolina at Chapel Hill, Chapel Hill, North Carolina, United States of America, 3 University of New Mexico, Albuquerque, New Mexico, United States of America, 4 RECOVER Community Representative, Bethesda, Maryland, United States of America, 5 New York University Grossman School of Medicine, New York, New York, United States of America, 6 Patient Led Research Collaborative, Calabasas, California, United States of America, 7 University of Virginia, Charlottesville, Virginia, United States of America, 8 Tulane University School of Public Health and Tropical Medicine, New Orleans, Louisiana, United States of America, 9 Johns Hopkins University School of Medicine, Public Health, and Nursing, Baltimore, Maryland, United States of America, 10 Emory University School of Medicine, Atlanta, Georgia, United States of America

◦ These authors contributed equally to this work.
* apreiss@rti.org

## Abstract

### Background

Preventing and treating post-acute sequelae of COVID-19 infection (PASC), commonly known as Long COVID, has become a public health priority. This study tests whether Paxlovid treatment in the acute phase of COVID-19 could help prevent the onset of PASC.

### Methods and findings

We used electronic health records from the National Clinical Cohort Collaborative to define a cohort of 445,738 patients who had COVID-19 since April 1, 2022, and were eligible for Paxlovid treatment due to risk for progression to severe COVID-19. We used the target trial emulation framework to estimate the effect of Paxlovid treatment on PASC incidence. We emulated a series of six sequential trials: one for each day of a 5-day treatment grace period. For each sequential trial, the treatment group was defined as patients prescribed Paxlovid on the trial start day, and the control group was defined as all patients meeting eligibility criteria who remained untreated on the trial start day. We pooled individual record-level data from the sequential trials for analysis. The follow-up period was 180 days. The primary outcome was overall PASC

which permits unrestricted use, distribution, and reproduction in any medium, provided the original author and source are credited.

**Data availability statement:** All data and code are available in the N3C Data Enclave to those with an approved protocol and data use request from an institutional review board. Data access is governed under the authority of the National Institutes of Health; more information on accessing the data can be found at https:// covid.cd2h.org/for-researchers. The Data Use Request ID for this study is RP-5677B5. This study used data from RECOVER release v187. See Haendel et. al. for additional detail on how data is ingested, managed, and protected within the N3C Data Enclave [https://doi.org/10.1093/ jamia/ocaa196]. All code was written for use in the Enclave on the Palantir Foundry platform [https://www.palantir.com/platforms/found-ry/], where the analysis can be reproduced by researchers [62]. It can be exported for review upon request.

**Funding:** This study was funded by the National Institutes of Health (NIH) National Center for Advancing Translational Sciences (NCATS) (OTA OT2HL161847 to all authors). NCATS contributed to the design, maintenance, and security of the N3C Data Enclave, and the NIH RECOVER Initiative, which coordinates the efforts of funded EHR and real-world data researchers in selecting and answering pressing Long COVID research questions. The funder of the study had no role in study design, data analysis, data interpretation, writing of the report, or the decision to submit the manuscript for publication.

**Competing interests:** The authors have declared that no competing interests exist.

**Abbreviations:** ARD, absolute risk difference; CCI, Charlson Comorbidity Index; CDC, Centers for Disease Control and Prevention; EHR, electronic health record; GBD, Global Burden of Disease; HR, hazard ratio; IPC, inverse probability of censoring; IPD-MA, individual patient data meta-analysis; IPT, inverse probability of treatment; N3C, National Clinical Cohort Collaborative; NNT, number needed to treat; PASC, post-acute sequelae of COVID-19; RECOVER, Researching COVID to Enhance Recovery; RR, relative risk; STROBE, strengthening the reporting of observational studies in epidemiology; TTE, target trial emulation; VA, United States Department of Veterans Affairs.

incidence measured using a computable phenotype. Secondary outcomes were incident cognitive, fatigue, and respiratory symptoms in the post-acute period. We controlled for a wide range of demographic and medical history covariates. Compared to the control group, Paxlovid treatment did not have a significant effect on overall PASC incidence or incident respiratory symptoms. It had a small protective effect against cognitive (relative risk [RR] 0.91; 95% CI [0.84, 0.98]; $p = 0.019$) and fatigue (RR 0.94; 95% CI [0.90, 0.98]; $p = 0.002$) symptoms. Finally, we estimated Paxlovid's effect on overall PASC incidence across strata of age, COVID-19 vaccination status, and Charlson Comorbidity Index (CCI) prior to COVID-19. We found small protective effects among patients aged 65 years or more (RR 0.92; 95% CI [0.88, 0.97]; $p < 0.001$; absolute risk difference [ARD] −0.43%; number needed to treat [NNT] 233) and with a CCI of 3 or 4 (RR 0.83; 95% CI [0.75, 0.92]; $p < 0.001$; ARD −1.30%; NNT 76). This study's main limitation is that the causal interpretation relies on the assumption that we controlled for all confounding variables.

## Conclusions

Although some prior observational studies suggested that Paxlovid held promise as a PASC preventive, this study—with a large, nationally sampled cohort; a contemporary study period; and causal inference methodology—found that Paxlovid treatment during acute COVID-19 had no effect on subsequent PASC incidence. Stratified analyses suggest that Paxlovid may have a small protective effect among higher-risk patients, but the NNT is high. In conclusion, we see Paxlovid as unlikely to become a definitive solution for PASC prevention.

## Author summary

### Why was this study done?

- Paxlovid is indicated to prevent severe COVID-19.

- Long COVID is more likely after more severe COVID-19, so there is a plausible mechanism for Paxlovid to reduce the risk of developing Long COVID by preventing severe COVID-19.

- Only a few studies have examined the relationship between Paxlovid and Long COVID, with mixed results.

- If Paxlovid helps prevent Long COVID, it could be a powerful addition to the public health effort to reduce the burden of COVID-19.

### What did the researchers do and find?

- We used a cohort of 445,738 patients from the National Clinical Cohort Collaborative's electronic health record database to estimate the effect of Paxlovid treatment during acute COVID-19 on the likelihood of developing Long COVID.

- We used the target trial emulation technique to estimate the causal effect of Paxlovid treatment using observational data.

- We found that Paxlovid treatment does not reduce the risk of Long COVID incidence.

- We found that Paxlovid treatment had a slightly stronger effect on certain symptoms (cognitive and fatigue) of Long COVID.

- We studied this effect separately across patients grouped by their age and medical history. We found that Paxlovid reduced Long COVID risk among older and sicker patients, but only by a small amount.

### *What do these findings mean?*

- Paxlovid is unlikely to become a definitive solution for preventing Long COVID.

- Although it might help a little among higher-risk people (older and sicker), the effect is not very big. For example, doctors would have to prescribe Paxlovid to 233 more people aged 65 years or more to prevent one case of PASC.

- Paxlovid may have a slightly stronger effect on certain Long COVID symptoms (cognitive and fatigue).

- This study's main limitation is that it estimates causal effects, but it is not randomized like a clinical trial. Instead, we control for other variables that could bias the estimate, but if we missed some important variables, the estimates could be incorrect.

## Introduction

Post-acute sequelae of COVID-19 infection (PASC), commonly known as Long COVID, affects people from all walks of life. Many people with PASC continue to feel the impacts of the disease years after infection. Mechanisms causing PASC remain largely unknown, and we have yet to identify a treatment that is consistently effective across the array of PASC manifestations. Therefore, developing effective PASC prevention strategies will be crucial to alleviating the long-term public health impact of COVID-19.

Nirmatrelvir with ritonavir (Paxlovid) was given an emergency use authorization in the United States in December 2021 for the treatment of patients with mild-to-moderate COVID-19 who are at high-risk for progression to severe COVID-19. Paxlovid has proven effective at preventing severe COVID-19, hospitalization, and death, with supporting evidence from clinical trials and real-world evidence [1–7].

In 2022−2023, several teams published case reports where Paxlovid was used to treat extant PASC [8–11]. This evidence motivated several clinical trials, including RECOVER-VITAL, to evaluate Paxlovid as a potential treatment for PASC [12]. Results of smaller trials have begun to emerge. In Stanford University's STOP-PASC trial, which included 155 participants, Paxlovid did not show benefit in improving extant fatigue, brain fog, body aches, cardiovascular symptoms, shortness of breath, or gastrointestinal symptoms [13].

In addition to treating PASC, researchers have begun to explore whether Paxlovid treatment in the acute phase of COVID-19 could help prevent the onset of PASC. One plausible pathway could be reducing infection severity. Several studies have found that more severe acute infection or hospitalization is associated with a higher-risk of PASC [14–17]. To our knowledge, few studies have explored Paxlovid as a PASC preventive, and results are mixed. A large, observational study from the US Department of Veterans Affairs (VA) found that Paxlovid treatment during the acute phase of COVID-19 was associated with a lower-risk of PASC, with a hazard ratio (HR) of 0.74 [18]. However, a target trial emulation (TTE), also using VA data, found Paxlovid had no effect on 30 of 31 post-COVID-19 conditions [19]. Two other smaller studies found that Paxlovid treatment was not associated with a reduced risk of PASC [20,21]. Because there is still no consensus definition of PASC, these studies used different outcome measures. In sum, the relationship between Paxlovid treatment and PASC onset remains uncertain.

At the time of writing, the PANORAMIC trial in the United Kingdom and the CanTreatCOVID trial in Canada have completed recruitment for arms which will receive Paxlovid during acute COVID-19 [22,23]. The PANORAMIC trial will focus on acute outcomes, but the CanTreatCOVID trial will include follow-up at 90 days and 36 weeks. CanTreatCOVID will provide valuable insight to the relationship between Paxlovid treatment and PASC onset. This study will provide real-world evidence to complement the findings from CanTreatCOVID and, hopefully, additional future trials.

Through the National Institute of Health's National Clinical Cohort Collaborative (N3C), and as part of the Researching COVID to Enhance Recovery (RECOVER) Initiative's electronic health record (EHR) data team, we have the opportunity to study Paxlovid as a PASC preventive using a large, nationally sampled cohort and an up-to-date study period consisting mostly of Omicron BA and later subvariant infections [24,25]. All analyses described here were performed within the secure N3C Data Enclave, which integrates EHR data for 21 million patients from over 230 data partners across the United States. N3C's methods for data acquisition, ingestion, and harmonization have been reported elsewhere [24,26,27]. This study used the TTE framework to estimate the effect of Paxlovid treatment in the acute phase of COVID-19 infection on the cumulative incidence of PASC among a cohort of patients eligible for Paxlovid treatment (i.e., with one or more risk factors for developing severe COVID-19) [28]. We hypothesized that Paxlovid treatment reduces the likelihood of PASC incidence.

## Methods

### Target trial emulation

We followed the two-step process for emulating target trials with observational data suggested by Hernán and Robins [29]. First, we articulated the causal question of interest in the form of a hypothetical trial protocol. Second, we emulated each component of this protocol using observational EHR data. We emulated a target randomized controlled trial of Paxlovid versus no treatment among adults with a COVID-19 index (defined as either a COVID-19 diagnosis [ICD-10 code U07.1] or a positive SARS-CoV-2 test result) between April 1, 2022 and February 28, 2023, and at least one risk factor for progression to severe acute COVID-19. Because Paxlovid is indicated for use within 5 days of symptom onset, the target trial included a treatment grace period of 5 days. To account for this grace period without introducing immortal time bias, we emulated six sequential trials, in which time zero ($t_0$) ranged from the COVID-19 index date ($t_{index}$) to $t_{index + 5}$. Each sequential trial compared patients who were prescribed Paxlovid on $t_0$ (treatment) to patients who were not (control). Patients were followed for 180 days after $t_{index}$. See the extended methods in S1 Text for details on sequential trial methods.

### Eligibility

Inclusion criteria were: (1) having a documented $t_{index}$ within the study period, (2) being ≥ 18 years of age at $t_{index}$ (due to potential differences in clinical characteristics and prescription practices between pediatric and adult patients [30,31]), and (3) having ≥1 risk factor for severe COVID-19 per Centers for Disease Control and Prevention (CDC) guidelines (age ≥50 years or diagnosis of a comorbidity associated with higher-risk of severe COVID-19 [32]). Baseline exclusion criteria were: (1) being hospitalized on $t_{index}$, (2) having prior history of PASC, and (3) being prescribed a drug with a severe interaction with Paxlovid in the 30 days prior to $t_{index}$ [33]. To ensure that data were captured from sites with high fidelity and adequate coverage, we also excluded sites with fewer than 500 or 5% of eligible patients treated with Paxlovid during the study period. Finally, we reassessed sequential exclusion criteria at $t_0$ for each sequential trial: (1) having died between $t_{index}$ and $t_0$, (2) having been hospitalized between $t_{index}$ and $t_0$, (3) having received a Paxlovid prescription between $t_{index}$ and $t_{0-1}$, and (4) having appeared in the same treatment group in a previous sequential trial. The final criterion prevents the control group from ballooning due to repeated inclusion of patients, which would increase computational expense while contributing little additional information.

## Outcome

We measured overall PASC incidence using a machine learning-based computable phenotype model, which gathers data for each patient in overlapping 100-day periods that progress through time, and issues a probability of PASC for each 100-day period [34]. The model was trained to classify whether patients have a U09.9 ("Post COVID-19 Condition") ICD-10 diagnosis code in each period, based on the patients' diagnoses during each period. The use of a computable phenotype is rare in the PASC literature. More often, researchers measure PASC using specific PASC diagnoses (U09.9) or they define a set of symptoms that constitute PASC and measure their incidence in the post-acute period. However, both of these measurements have problems that the computable phenotype avoids. PASC diagnoses are rare, and diagnosis is likely driven by access to care, which also affects the likelihood of Paxlovid treatment, leading to potential bias. The computable phenotype can help identify undiagnosed patients who have symptoms similar to those with U09.9 diagnoses. Measuring PASC using a specific set of symptoms can lead to false positives (symptoms with etiologies other than COVID-19) and false negatives (related symptoms not included in the definition). The computable phenotype can learn to avoid these errors. However, PASC is also a heterogeneous condition, so the use of symptom-specific outcomes is an important complement to the computable phenotype.

To measure PASC at a more granular level, we also measured incident PASC symptoms in the cognitive, fatigue, and respiratory clusters proposed by the Global Burden of Disease (GBD) Study ("GBD symptom clusters" henceforth) [35]. These clusters were the most frequently reported symptoms in a meta-analysis of Long COVID studies [35]. We estimated the effect of treatment on both individual symptom clusters, and a composite measure of any incident symptom across all clusters. See the extended methods in S1 Text for details on outcome definitions and rationale.

## Statistical analysis

To emulate the target trial from a series of sequential trials, we pooled patient-level records from each sequential trial. This approach is akin to one-stage individual patient data meta-analysis (IPD-MA), where the estimand of the sequential trial meta-analysis is equivalent to the estimand of the target trial [36]. However, unlike one-stage IPD-MA, it is not necessary to account for clustering of participants within each trial, since all participants come from the same underlying cohort. Also, unlike IPD-MA, weighting is necessary to establish exchangeability between treatment groups, because treatment is not randomly assigned within each trial.

We used stabilized inverse probability of treatment (IPT) weighting to emulate random assignment through exchangeability between treatment groups. For the pooled cohort, we used a single logistic regression model to estimate treatment propensity based on a set of baseline covariates. We selected covariates based on a theoretical causal model, shown in Fig A in S1 Text. Baseline covariates included: sex, age (binned), race and ethnicity, prior history of individual comorbid conditions captured in the Charlson Comorbidity Index (CCI), value of the composite CCI (binned), prior history of conditions associated with risk of severe COVID-19 (as defined by the CDC Paxlovid eligibility criteria [32]), Community Well-Being Index (binned), number of visits in the year prior to index (binned), number of hospitalizations in the year prior to index (binned), month of COVID-19 onset, and site of care provision. See the extended methods in S1 Text for detailed rationale for each covariate.

Because loss to follow-up was more common in the control group, we also used stabilized inverse probability of censoring (IPC) weighting to produce a pseudo-cohort in which censoring was random with respect to treatment. For the pooled cohort, we used a single logistic regression model to estimate censoring propensity based on the same set of baseline covariates and the treatment group. This approach treats the relative likelihood of censoring across groups as time-invariant, an assumption that we verified by examining the cumulative incidence of censoring by treatment group over time. Finally, we generated combined inverse probability weights as the product of IPT and IPC weights.

The estimand was the cumulative incidence of PASC from 29 to 180 days after COVID-19 index. To estimate the cumulative incidence of PASC, we used Aalen–Johansen estimators, where stabilized, trimmed, and combined IP weights

were used as time-fixed weights, and death was treated as a competing risk. We used bootstrapping with 500 iterations to estimate the 95% confidence interval at a two-sided alpha of 0.05. To estimate the average treatment effect, we took the relative and absolute difference in cumulative incidence in the treatment and control groups.

We followed patients for 180 days following their COVID-19 index date. Follow-up started at day 1, and we started observing the outcome at day 29 to avoid attributing acute symptoms to PASC. Computable phenotype PASC predictions and GBD symptoms between days 1 and 28 were ignored. Censorship was possible from days 1 to 180. We censored patients at the date of their last documented visit in the EHR. Death as a competing risk was also possible between days 1 and 180.

### Stratified analyses

We conducted stratified analyses for age, CCI, and COVID-19 vaccination status. We defined continuous strata with the same bins used for these covariates in the treatment and censoring models (age in years: 18–24, 25–34, 35–49, 50–64, 65+; CCI: 0, 1–2, 3–4, 5–10, 11+). For the vaccination-stratified analysis, we used the cohort subset from the vaccination subanalysis (see the following section) and defined unvaccinated and fully vaccinated strata (see S1 Text for details). For each stratum, we ran the full analysis pipeline independently, such that treatment and control groups in the stratum were exchangeable.

### Subanalyses and sensitivity analyses

We also conducted two subanalyses. The first used a "VA-like cohort" designed to mirror the study period used in Xie and colleagues (2023) and the demographics of VA patients [18]. It used an earlier study start date (including the first Omicron wave in early 2022) and only included male patients over 65 years old. The intent of this subanalysis was to minimize the differences between our studies and allow for more direct comparison with previously reported findings on this topic. The second subanalysis included COVID-19 vaccination status as an additional covariate, and was conducted in a subset of sites with high-quality vaccination data. We did not include vaccination as a covariate in the primary analysis, because it is subject to substantial measurement error in most EHRs. The intent of this subanalysis was to assess whether vaccination led to unmeasured confounding in the primary analysis. Finally, we conducted several sensitivity analyses to test sensitivity to estimation methods, inclusion and exclusion criteria, computable phenotype prediction threshold, COVID-19 index definition, and time period of outcome observation. See the extended methods in S1 Text for sensitivity analysis details.

### Ethics approval and consent to participate

The N3C data transfer to NCATS is performed under a Johns Hopkins University Reliance Protocol # IRB00249128 or individual site agreements with NIH. The N3C Data Enclave is managed under the authority of the NIH; information can be found at https://ncats.nih.gov/n3c/resources. The work was performed under DUR RP-5677B5. All results are reported in adherence with the Strengthening the Reporting of Observational Studies in Epidemiology (STROBE) guidelines (see S1 STROBE Checklist) [37].

## Results

### Patient characteristics

The study cohort included 445,738 patients, of whom 151,180 (33.92%) had a Paxlovid prescription within the treatment grace period, and 18,663 (4.20%) had PASC (U09.9 diagnosis or computable phenotype prediction over 0.9 from 29 to 180 days after index). Among treated patients, 134,401 (88.90%) were prescribed Paxlovid on the same day as COVID-19 index, and 146,931 (97.19%) were prescribed Paxlovid within 1 day of COVID-19 index. The pooled sequential trial cohort included 460,803 patient records, of which 151,180 (32.81%) were in the treatment group (Due to the sequential trial design, patients who were treated on days 1–5 could appear in both the treatment and control groups).

During the study period, 208 (0.14%) patients treated with Paxlovid and 975 (0.33%) untreated patients died. A total of 4,341 (0.97%) patients had a post-acute symptom in the cognitive symptom cluster, 12,569 (2.82%) patients had a post-acute symptom in the fatigue symptom cluster, and 22,596 (5.07%) had a post-acute symptom in the respiratory symptom cluster. Among patients with a PASC diagnosis or computable phenotype prediction, 7.44% had a post-acute symptom in the cognitive symptom cluster, 19.82% had a post-acute symptom in the fatigue symptom cluster, and 35.71% had a post-acute symptom in the respiratory symptom cluster. A co-occurrence matrix, showing the percentage of patients with each outcome who also had other outcomes, is shown in Fig B in S1 Text. After applying the eligibility criteria to the patient population and study sites, a total of 28 of 36 study sites were retained. The CONSORT flow diagram is shown in Fig 1. The characteristics of all patients during the study period are presented in Table 1, stratified by treatment group. IPT weighting achieved balance across all covariates, as shown in Fig 2. The distribution of stabilized, trimmed, and combined IPT and IPC weights had a median of 0.86 and a standard deviation of 0.62. The target trial protocol and emulation approach are presented in Table 2.

## Effect of Paxlovid treatment on PASC incidence

We found that Paxlovid treatment during acute COVID-19 had no effect on overall PASC incidence or incident respiratory symptoms, and a small effect on incident cognitive and fatigue symptoms. Table 3 shows estimates of cumulative PASC incidence and treatment effects (relative risk [RR], absolute risk difference [ARD]) across all analyses, with number needed to treat [NNT] estimated for results that were statistically significant. Fig 3 shows RR for all analyses. Fig 4 shows cumulative incidence functions for the main analyses.

For overall PASC onset, measured by our PASC computable phenotype, adjusted cumulative incidence estimates were 4.53% (95% CI [4.40, 4.66]) for treated patients and 4.60% (95% CI [4.51, 4.68]) for untreated patients. The RR of PASC was 0.99 (95% CI [0.95, 1.02]; $p = 0.43$), with an ARD of −0.10% (95% CI [−0.20%, 0.10%]; $p = 0.41$). The RR of any GBD symptom was 0.99 (95% CI [0.96, 1.01]; $p = 0.27$), with an ARD of −0.10% (95% CI [−0.30%, 0.10%]; $p = 0.27$). For the cognitive symptom cluster, RR was 0.91 (95% CI [0.84, 0.98]; $p = 0.019$), with an ARD of −0.10% (95% CI [−0.20%, 0.00%]; $p = 0.019$) and NNT of 988. For the fatigue symptom cluster, RR was 0.94 (95% CI [0.90, 0.98]; $p = 0.002$), with an ARD of −0.20% (95% CI [−0.30%, −0.10%]; $p = 0.002$) and NNT of 508. For the respiratory symptom cluster, RR was 1.01 (95% CI [0.98, 1.05]; $p = 0.48$), with an ARD of 0.10% (95% CI [−0.10%, 0.20%]; $p = 0.47$).

## Stratified analyses

Among the age strata, Paxlovid had a protective effect on overall PASC onset only among patients aged 65 years or more. The RR in this group was 0.92 (95% CI [0.88, 0.97]; $p < 0.001$), with an ARD of −0.43% (95% CI [−0.70%, −0.20%]; $p < 0.001$) and NNT of 233. Paxlovid had an anti-protective effect among patients aged 18–24 years (RR = 1.31, 95% CI [1.06, 1.62]; $p = 0.03$) and among patients aged 35–49 years (RR = 1.08, 95% CI [1.01, 1.16]; $p = 0.03$). The proportion of patients treated increased monotonically by ascending age group (18–24 years: 14.47%, 25–34 years: 19.80%, 35–49 years: 30.47%, 50–64 years: 37.99%, 65+ years: 44.14%).

Among the CCI strata, Paxlovid had a significant effect on overall PASC onset only among patients with a CCI of 3–4. The RR in this comorbidity group was 0.83 (95% CI [0.75, 0.92]; $p < 0.001$), with an ARD of −1.30% (95% CI [−2.00%, −0.60%]; $p < 0.001$) and NNT of 76. There was no significant effect among patients with a CCI of 5–10 (RR = 0.91, 95% CI [0.78, 1.07]; $p = 0.24$). Although the magnitude of the effect in this group was similar, statistical power was lower, and the result was not significant. Similarly, there was no significant effect among patients with a CCI of 11 or greater (RR = 1.40, 95% CI [0.78, 2.50]; $p = 0.26$), though the statistical power was very limited in this stratum. The proportion of patients treated varied less by CCI than by age group, and was non-monotonic. It ranged from 33.62% among patients with a CCI of 0 to 39.88% among patients with a CCI of 1–2.

Among the COVID-19 vaccination status strata, Paxlovid's effect on overall PASC onset did not vary. Its effect was not significant among both unvaccinated and fully vaccinated patients.

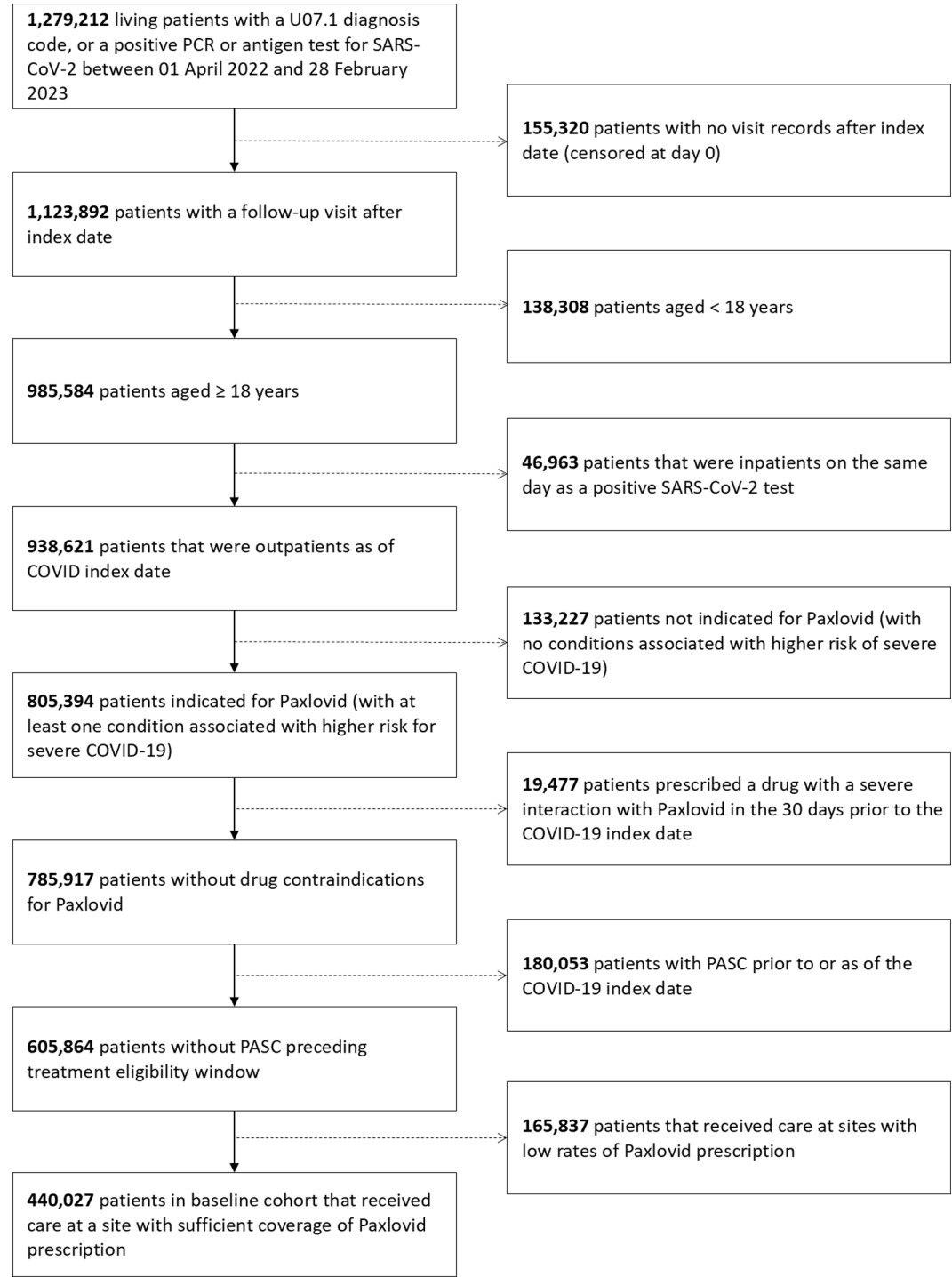

**Fig 1. CONSORT diagram: Study cohort and flow of emulated trial.** U07.1, COVID-19 ICD-10 code; PCR, polymerase chain reaction; PASC, post-acute sequelae of COVID-19.

## Subanalyses

**Table 1. Descriptive population characteristics within the National Clinical Cohort Collaborative Cohort. Each cell shows the number and percentage of patients in the treatment group with the characteristic.**

| Characteristic | Treatment group | |
|---|---|---|
| | No Paxlovid [N, (%)] | Paxlovid [N, (%)] |
| | (N = 294,558) | (N = 151,180) |
| Outcomes | | |
| Computable phenotype prediction or diagnosis[1] (29–180 days) | 11,955 (4.1%) | 6,708 (4.4%) |
| Cognitive symptom cluster (29–180 days) | 3,057 (1.0%) | 1,183 (1.0%) |
| Fatigue symptom cluster (29–180 days) | 8,533 (2.9%) | 3,366 (2.8%) |
| Respiratory symptom cluster (29–180 days) | 14,879 (5.1%) | 7,717 (5.1%) |
| Death (1–28 days) | 366 (0.1%) | 51 (0.0%) |
| Death (29–180 days) | 609 (0.2%) | 157 (0.1%) |
| Sex | | |
| Female | 182,354 (61.9%) | 90,390* (59.8%) |
| Male | 112,180 (38.1%) | 60,780* (40.2%) |
| Missing | 24 (0.0%) | < 20 (0.0%) |
| Age (in years) | | |
| 18–24 | 20,980 (7.1%) | 3,521 (2.3%) |
| 25–34 | 43,521 (14.8%) | 10,622 (7.0%) |
| 35–49 | 60,620 (20.6%) | 25,908 (17.1%) |
| 50–64 | 80,576 (27.4%) | 47,051 (31.1%) |
| 65+ | 88,861 (30.2%) | 64,078 (42.4%) |
| Race and Ethnicity | | |
| Asian Non-Hispanic | 12,205 (4.1%) | 5,327 (3.5%) |
| Black Non-Hispanic | 39,604 (13.4%) | 13,387 (8.9%) |
| Hispanic or Latino any race | 21,782 (7.4%) | 7,659 (5.1%) |
| White Non-Hispanic | 202,346 (68.7%) | 116,890 (77.3%) |
| Other Non-Hispanic | 5,007 (1.7%) | 1,466 (1.0%) |
| Unknown | 13,614 (4.6%) | 6,451 (4.3%) |
| Charlson Comorbidity Index | | |
| 0 | 170,233 (57.8%) | 83,256 (55.1%) |
| 1–2 | 73,299 (24.9%) | 45,655 (30.2%) |
| 3–4 | 18,150 (6.2%) | 10,542 (7.0%) |
| 5–10 | 7,142 (2.4%) | 3,781 (2.5%) |
| 11+ | 332 (0.1%) | 195 (0.1%) |
| Missing | 25,402 (8.6%) | 7,751 (5.1%) |
| Number of visits in prior year | | |
| 0 | 35,440 (12.0%) | 11,760 (7.8%) |
| 1–3 | 61,873 (21.0%) | 22,729 (15.0%) |
| 4–9 | 72,178 (24.5%) | 38,032 (25.2%) |
| 10–20 | 69,786 (23.7%) | 44,641 (29.5%) |
| >20 | 55,281 (18.8%) | 34,018 (22.5%) |
| Number of hospitalizations in prior year | | |
| 0 | 281,817 (95.7%) | 145,268 (96.1%) |
| 1 | 10,997 (3.7%) | 5,212 (3.4%) |
| >1 | 1,744 (0.6%) | 700 (0.5%) |
| Community Wellbeing Index[2] | | |
| 0–45 | 1,993 (0.7%) | 748 (0.5%) |

*(Continued)*

**Table 1.** (Continued)

| Characteristic | Treatment group | |
|---|---|---|
| | No Paxlovid [N, (%)] | Paxlovid [N, (%)] |
| | (N = 294,558) | (N = 151,180) |
| 46–55 | 112,150 (38.1%) | 48,167 (31.9%) |
| 56–65 | 130,428 (44.3%) | 71,043 (47.0%) |
| 65+ | 13,148 (4.5%) | 11,724 (7.8%) |
| Missing | 36,839 (12.5%) | 19,498 (12.9%) |
| Censoring events | | |
| Lost to follow-up (no further visits in EHR) (1–28 days) | 11,246 (3.8%) | 2,714 (1.8%) |
| Lost to follow-up (no further visits in EHR) (29–180 days) | 25,508 (8.7%) | 8,747 (5.8%) |
| Month of COVID-19 diagnosis | | |
| April 2022 | 17,343 (5.9%) | 4,683 (3.1%) |
| May 2022 | 39,544 (13.4%) | 16,227 (10.7%) |
| June 2022 | 40,533 (13.8%) | 18,749 (12.4%) |
| July 2022 | 44,303 (15.0%) | 22,465 (14.9%) |
| August 2022 | 36,798 (12.5%) | 18,256 (12.1%) |
| September 2022 | 23,473 (8.0%) | 12,512 (8.3%) |
| October 2022 | 18,304 (6.2%) | 9,551 (6.3%) |
| November 2022 | 18,031 (6.1%) | 10,764 (7.1%) |
| December 2022 | 24,463 (8.3%) | 17,157 (11.3%) |
| January 2023 | 19,090 (6.5%) | 11,566 (7.7%) |
| February 2023 | 12,676 (4.3%) | 9,250 (6.1%) |

[1]Any PASC (CP or U09.9) between 28 days following a positive SARS-CoV-2 test result to 180 days post-index.

[2]CWBI is a measure of five interrelated community-level domains: Healthcare access (ratios of healthcare providers to population), Resource access (libraries and religious institutions, employment, and grocery stores), Food access (access to grocery stores and produce), Housing and transportation (home values, ratio of home value to income, and public transit use), and Economic security (rates of employment, labor force participation, health insurance coverage rate, and household income above the poverty level) [63].

Abbreviation: EHR, electronic health record.

The VA-like cohort included 73,212 male patients 65 years or older with a COVID-19 index between January 3, 2022 and December 31, 2022 [18]. Of this cohort, 26,103 (35.65%) were treated with Paxlovid. The RR for overall PASC (computable phenotype or U09.9 diagnosis) was 0.99 (95% CI [0.91, 1.08]; $p = 0.77$). For the cognitive, fatigue, and respiratory symptom clusters, RR estimates were 0.90 (95% CI [0.79, 1.04]; $p = 0.15$), 0.93 (95% CI [0.84, 1.03]; $p = 0.15$), and 0.92 (95% CI [0.86, 0.99]; $p = 0.03$), respectively. Cumulative incidence functions are shown in Fig C in S1 Text.

The vaccination-aware cohort included 115,823 patients from eight sites that met vaccination data quality criteria. Of this cohort, 91,676 (79.15%) were fully vaccinated prior to index, 52,893 (45.67%) were treated with Paxlovid, and 4,746 (4.10%) had PASC. Among fully vaccinated patients, 45,115 (49.21%) were treated, as compared to 7,778 (32.21%) unvaccinated patients. The RR for overall PASC was 1.00 (95% CI [0.94, 1.06]; $p = 0.96$). For the cognitive, fatigue, and respiratory symptom clusters, RR estimates were 0.90 (95% CI [0.78, 1.03]; $p = 0.13$), 0.99 (95% CI [0.92, 1.07]; $p = 0.83$), and 0.97 (95% CI [0.92, 1.03]; $p = 0.34$), respectively. Cumulative incidence functions are shown in Fig D in S1 Text.

Our findings were not sensitive to the following: treating Paxlovid prescriptions without an accompanying U07.1 diagnosis or lab test as COVID-19 index events; treating death as a censoring event instead of a competing risk; using a doubly robust estimator and the HR of Paxlovid treatment as the estimand; varying the computable phenotype prediction threshold; varying the period of PASC observation; assuming that patients did not get PASC during the period after their last documented visit (i.e., not censoring on last visit date); and excluding patients who received Molnupiravir or Ritonavir

PLOS Medicine

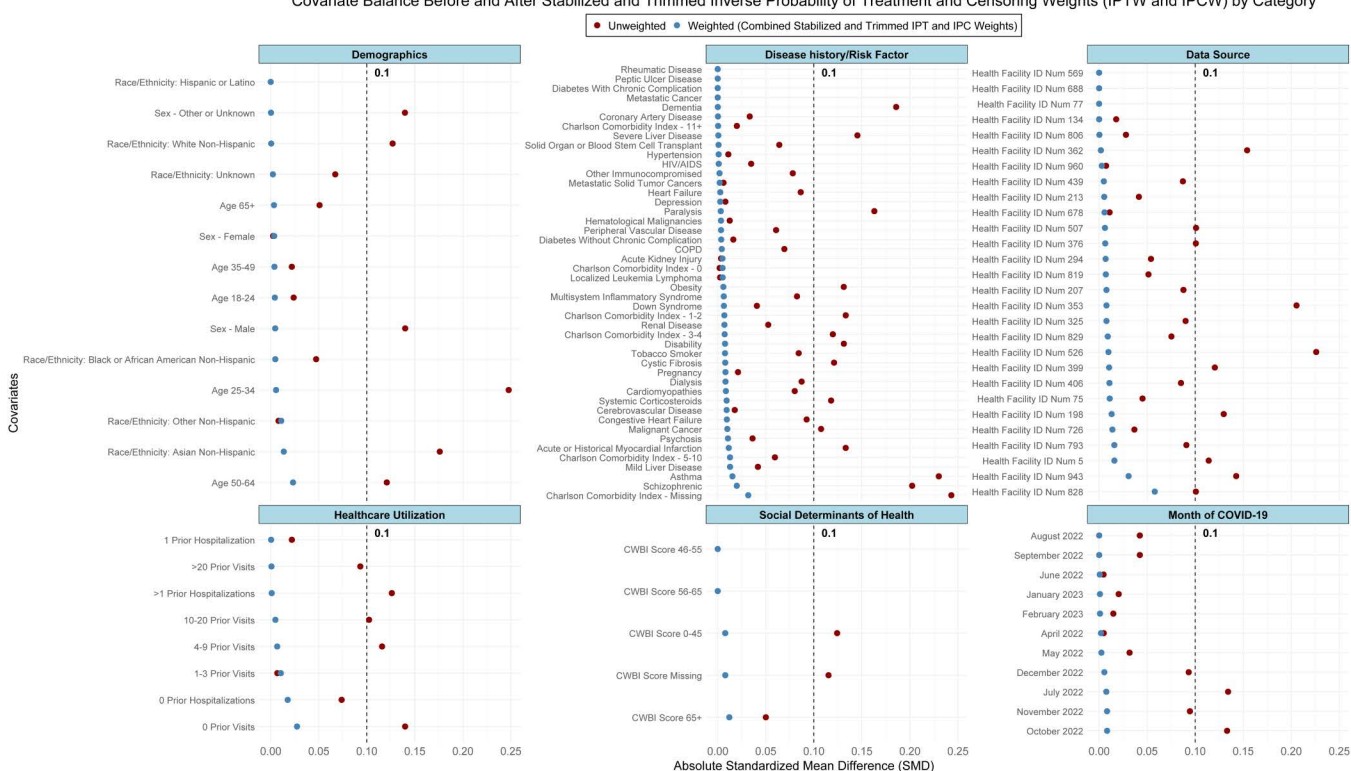

**Fig 2. Covariate balance before and after stabilized and trimmed inverse probability of censoring and treatment weighting.** IPCW, inverse probability of censoring weighting; IPTW, inverse probability of treatment weighting; CWBI, community well-being index.

during the 5-day treatment grace period. When U07.1 diagnoses without an accompanying lab test were not treated as COVID-19 index events, we found a small anti-protective effect. Thus, our findings were sensitive in this respect to the definition of a COVID-19 index event, but the resulting conclusion was not.

## Discussion

In this target trial emulation using the N3C database and a nationally sampled cohort of patients eligible for Paxlovid treatment (i.e., with one or more risk factors for severe COVID-19), we found that Paxlovid treatment during the acute phase of COVID-19 did not have an effect on overall PASC incidence (RR = 0.99; ARD = −0.10%). Paxlovid also had no significant effect on incident respiratory symptoms, though it had a small but statistically significant protective effect on incident cognitive symptoms (RR = 0.91; ARD = −0.10%; NNT = 988) and fatigue symptoms (RR = 0.94; ARD = −0.20%; NNT = 508).

Differing effects by symptom cluster suggest that Paxlovid may have more impact on the underlying causes of certain symptoms. In the literature, multiple PASC etiologies have been proposed. The chief hypotheses are that, relative to healthy convalescents, those with PASC may be experiencing (1) an aberrant autoimmune response triggered by the virus, (2) organ, tissue, or vascular dysfunction related to inflammatory cascades following infection, and/or (3) persistent viremia due to increased viral load or viral reservoirs. We do not yet know which symptoms are caused by which mechanisms. Paxlovid treatment decreases viral load, and thus could plausibly have more impact on symptoms arising from the third factor [38]. Our findings allow us to generate the hypothesis that viral load may be a more common cause of cognitive and fatigue symptoms than of other PASC symptoms.

**Table 2. Protocol of a target trial emulation to estimate the effect of Paxlovid treatment during acute COVID-19 on cumulative PASC incidence.**

| Protocol component | Description under target trial conditions | Method of target trial emulation |
|---|---|---|
| Eligibility criteria | Persons aged 18 and older, with no history of PASC, who are not currently hospitalized, who have an acute COVID-19 infection, who are eligible for Paxlovid treatment due to the presence of one or more risk factors for severe COVID-19 as per CDC guidelines [32], and who are taking no medications with contraindications to Paxlovid. | Same, with acute COVID-19 infection defined as a COVID-19 index (either a documented COVID-19 diagnosis or positive SARS-CoV-2 lab test), and with drug-drug contraindications assessed in the 30 days prior to COVID-19 index. Additionally, for each sequential trial (see below), reassess criteria that persons must be alive, unhospitalized, have not received a Paxlovid prescription between COVID-19 index and the day before the start date of the sequential trial, and have not appeared in the same treatment group of a previous sequential trial. |
| Treatment strategies | Treatment arm: Paxlovid prescribed within a five-day grace period after the date the patient presented with acute COVID-19. Adherence was not monitored in accordance with the goal of estimating an intention-to-treat effect. Standard of care followed in all other respects, including the potential prescription of additional doses of Paxlovid. Control arm: No treatment. Standard of care followed in all other respects, including the potential prescription of Paxlovid after the five-day grace period. | For each day $d$ of a five-day grace period after COVID-19 index, emulate a sequential trial, with treatment and control groups defined as follows: Treatment group: Paxlovid prescribed on day $d$ (indicated by a Paxlovid or nirmatrelvir drug exposure record in the EHR). Control group: All patients meeting eligibility criteria and not meeting the treatment group definition. |
| Assignment procedures | Participants will be randomly assigned to treatment or control arm at the date they present with acute COVID-19 and will be aware of their treatment assignment. | Patients from all sequential trials will be pooled into a single cohort and assigned weights based on treatment propensity scores to ensure exchangeability of treatment and control groups and emulate random assignment conditional on measured variables. |
| Follow-up period | Each patient will be followed for 180 days after treatment. | Patients will be censored at 180 days after COVID-19 index or at the time of their last recorded visit in the EHR, whichever is earlier. To control for informative censoring due to different rates of loss to follow-up across treatment groups, patients will be assigned weights based on censoring propensity scores. |
| Outcome | Clinical diagnosis of PASC within follow-up period | Primary outcome: Clinical diagnosis of PASC or computable phenotype predicted probability of PASC >0.9 between 29 and 180 days after COVID-19 index. Secondary outcomes: Incident cognitive, fatigue, or respiratory symptoms between 29 and 180 days after COVID-19 index (with *incident* defined as symptoms that did not occur in the three years prior to COVID-19 index). |
| Causal contrasts | Intention-to-treat effect | Intention-to-treat effect |
| Analysis plan | Measure the relative risk of PASC diagnosis across treatment arms. | Estimate cumulative incidence of PASC in each treatment arm using Aalen-Johansen estimators weighted for treatment and censoring propensity and with death as a competing risk; estimate relative risk based on point estimates and variances of cumulative incidence estimates. |

Abbreviations: CDC, United States Centers for Disease Control and Prevention; EHR, electronic health record; PASC, post-acute sequelae of COVID-19.

In the age-stratified analysis, we found that Paxlovid had a significant protective effect on overall PASC incidence only among patients aged 65 years or more (RR = 0.92; ARD = −0.43%; NNT = 233). We also found that Paxlovid had an anti-protective effect among two younger age groups (18–24 years and 35–49 years) but no effect on the age group between them (25–34 years), although the confidence intervals for all these age groups overlap. We know of no mechanism through which Paxlovid could increase the risk of PASC. A more likely explanation is unmeasured confounding particular to these strata. Younger patients may be less motivated to seek treatment for COVID-19, and providers may be less likely to prescribe Paxlovid to younger patients. An acute infection would need to be especially severe for such a patient to obtain Paxlovid. This speculative interpretation is supported by lower treatment rates among younger patients (e.g., 14.47% among patients aged 18–24 years, 44.14% among patients aged 65 years or more).

**Table 3. Cumulative incidence and Absolute Risk Difference estimates across all analyses.**

| Analysis | Cumulative incidence (95% CI) | | Relative risk (95% CI) | p-value | Absolute Risk Difference (95% CI) | p-value | NNT |
|---|---|---|---|---|---|---|---|
| | Paxlovid | No Paxlovid | | | | | |
| **Main Results** | | | | | | | |
| Computable phenotype PASC prediction or U09.9 diagnosis | 0.045 (0.044, 0.047) | 0.046 (0.045, 0.047) | 0.986 (0.953, 1.021) | 0.426 | −0.001 (−0.002, 0.001) | 0.414 | – |
| Cognitive Symptom Cluster | 0.010 (0.009, 0.011) | 0.011 (0.011, 0.012) | 0.909 (0.839, 0.984) | 0.019 | −0.001 (−0.002, 0.000) | 0.017 | 988 |
| Fatigue Symptom Cluster | 0.029 (0.028, 0.030) | 0.031 (0.030, 0.032) | 0.937 (0.898, 0.977) | 0.002 | −0.002 (−0.003, −0.001) | 0.002 | 508 |
| Respiratory Symptom Cluster | 0.055 (0.053, 0.056) | 0.054 (0.053, 0.055) | 1.012 (0.980, 1.045) | 0.476 | 0.001 (−0.001, 0.002) | 0.472 | – |
| Any Symptom Cluster | 0.086 (0.084, 0.088) | 0.087 (0.086, 0.088) | 0.986 (0.962, 1.011) | 0.271 | −0.001 (−0.003, 0.001) | 0.269 | – |
| **Subanalyses** | | | | | | | |
| VA-like Cohort CP prediction or U09.9 diagnosis | 0.049 (0.046, 0.053) | 0.050 (0.048, 0.052) | 0.987 (0.907, 1.075) | 0.767 | −0.001 (−0.005, 0.004) | 0.766 | – |
| VA-like Cohort Cognitive Symptom Cluster | 0.017 (0.015, 0.019) | 0.018 (0.017, 0.020) | 0.903 (0.787, 1.036) | 0.145 | −0.002 (−0.004, 0.001) | 0.135 | – |
| VA-like Cohort Fatigue Symptom Cluster | 0.035 (0.032, 0.038) | 0.038 (0.036, 0.040) | 0.929 (0.840, 1.027) | 0.151 | −0.003 (−0.006, 0.001) | 0.144 | – |
| VA-like Cohort Respiratory Symptom Cluster | 0.058 (0.054, 0.061) | 0.062 (0.060, 0.065) | 0.923 (0.859, 0.993) | 0.031 | −0.005 (−0.009, −0.001) | 0.028 | 210 |
| VA-like Cohort Any Symptom Cluster | 0.097 (0.092, 0.102) | 0.104 (0.101, 0.107) | 0.929 (0.878, 0.982) | 0.009 | −0.007 (−0.013, −0.002) | 0.008 | 135 |
| Vaccination-Aware CP prediction or U09.9 diagnosis | 0.044 (0.042, 0.046) | 0.044 (0.042, 0.046) | 0.999 (0.943, 1.058) | 0.962 | 0.000 (−0.003, 0.002) | 0.962 | – |
| Vaccination-Aware Cognitive Symptom Cluster | 0.010 (0.009, 0.011) | 0.011 (0.010, 0.012) | 0.900 (0.784, 1.033) | 0.133 | −0.001 (−0.002, 0.000) | 0.131 | – |
| Vaccination-Aware Fatigue Symptom Cluster | 0.029 (0.027, 0.030) | 0.029 (0.028, 0.030) | 0.992 (0.922, 1.067) | 0.833 | 0.000 (−0.002, 0.002) | 0.833 | – |
| Vaccination-Aware Respiratory Symptom Cluster | 0.053 (0.051, 0.056) | 0.055 (0.053, 0.057) | 0.974 (0.924, 1.027) | 0.336 | −0.001 (−0.004, 0.001) | 0.335 | – |
| Vaccination-Aware Any Symptom Cluster | 0.084 (0.081, 0.087) | 0.086 (0.084, 0.088) | 0.979 (0.937, 1.022) | 0.326 | −0.002 (−0.005, 0.002) | 0.325 | – |
| **Stratified Analyses** | | | | | | | |
| Age 18–24 | 0.031 (0.025, 0.038) | 0.024 (0.023, 0.025) | 1.307 (1.056, 1.618) | 0.014 | 0.007 (0.001, 0.014) | 0.030 | 136 |
| Age 25–34 | 0.032 (0.028, 0.036) | 0.031 (0.030, 0.032) | 1.025 (0.899, 1.168) | 0.713 | 0.001 (−0.003, 0.005) | 0.716 | – |
| Age 35–49 | 0.046 (0.043, 0.049) | 0.042 (0.042, 0.043) | 1.081 (1.009, 1.160) | 0.028 | 0.003 (0.000, 0.007) | 0.033 | – |
| Age 50–64 | 0.044 (0.042, 0.046) | 0.044 (0.043, 0.045) | 1.003 (0.950, 1.058) | 0.921 | 0.000 (−0.002, 0.002) | 0.921 | – |
| Age 65+ | 0.052 (0.050, 0.054) | 0.056 (0.055, 0.057) | 0.924 (0.883, 0.966) | <0.001 | −0.004 (−0.007, −0.002) | <0.001 | 233 |
| CCI 0 | 0.036 (0.034, 0.037) | 0.035 (0.035, 0.036) | 1.012 (0.966, 1.060) | 0.628 | 0.000 (−0.001, 0.002) | 0.63 | – |
| CCI 1–2 | 0.056 (0.053, 0.059) | 0.056 (0.055, 0.057) | 1.005 (0.956, 1.056) | 0.845 | 0.000 (−0.003, 0.003) | 0.846 | – |
| CCI 3–4 | 0.066 (0.059, 0.072) | 0.079 (0.077, 0.081) | 0.833 (0.753, 0.922) | <0.001 | −0.013 (−0.020, −0.006) | <0.001 | 76 |
| CCI 5–10 | 0.082 (0.069, 0.094) | 0.090 (0.086, 0.093) | 0.910 (0.777, 1.066) | 0.244 | −0.008 (−0.021, 0.005) | 0.225 | – |
| CCI 11+ | 0.154 (0.067, 0.240) | 0.110 (0.095, 0.125) | 1.399 (0.782, 2.502) | 0.257 | 0.044 (−0.044, 0.132) | 0.33 | – |
| COVID-19 vaccination status: unvaccinated | 0.040 (0.034, 0.045) | 0.034 (0.033, 0.036) | 1.155 (1.000, 1.310) | 0.05 | 0.005 (0.000, 0.011) | 0.063 | – |
| COVID-19 vaccination status: fully vaccinated | 0.044 (0.042, 0.047) | 0.044 (0.043, 0.044) | 1.022 (0.966, 1.082) | 0.449 | 0.001 (−0.002, 0.003) | 0.453 | – |
| **Sensitivity Analyses** | | | | | | | |
| U09.9 Code Diagnosis | 0.004 (0.004, 0.005) | 0.004 (0.004, 0.004) | 1.112 (0.994, 1.243) | 0.064 | 0.000 (0.000, 0.001) | 0.070 | – |
| PASC Computable Phenotype Threshold—0.75 | 0.097 (0.095, 0.099) | 0.097 (0.096, 0.098) | 1.000 (0.976, 1.024) | 0.979 | 0.000 (−0.002, 0.002) | 0.979 | – |

*(Continued)*

**Table 3.** (Continued)

| Analysis | Cumulative incidence (95% CI) | | Relative risk (95% CI) | p-value | Absolute Risk Difference (95% CI) | p-value | NNT |
|---|---|---|---|---|---|---|---|
| | Paxlovid | No Paxlovid | | | | | |
| PASC Computable Phenotype Threshold—0.80 | 0.080 (0.078, 0.081) | 0.080 (0.079, 0.081) | 0.997 (0.972, 1.023) | 0.834 | 0.000 (−0.002, 0.002) | 0.834 | – |
| PASC Computable Phenotype Threshold—0.85 | 0.063 (0.061, 0.064) | 0.063 (0.062, 0.064) | 0.988 (0.959, 1.018) | 0.424 | −0.001 (−0.003, 0.001) | 0.423 | – |
| PASC Computable Phenotype Threshold—0.95 | 0.028 (0.027, 0.029) | 0.027 (0.027, 0.028) | 1.018 (0.973, 1.065) | 0.439 | 0.000 (−0.001, 0.002) | 0.441 | – |
| Paxlovid Treatment as Index Event | 0.045 (0.044, 0.047) | 0.046 (0.045, 0.047) | 0.988 (0.958, 1.019) | 0.443 | −0.001 (−0.002, 0.001) | 0.442 | – |
| Positive Lab-only Index Events | 0.045 (0.042, 0.047) | 0.041 (0.040, 0.042) | 1.085 (1.021, 1.153) | 0.008 | 0.004 (0.001, 0.006) | 0.010 | 285 |
| CP prediction or U09.9 diagnosis (29–365 days) | 0.097 (0.095, 0.100) | 0.097 (0.096, 0.099) | 1.000 (0.974, 1.027) | 0.982 | 0.000 (−0.003, 0.003) | 0.982 | – |
| CP prediction or U09.9 diagnosis (90–180 days) | 0.035 (0.034, 0.036) | 0.035 (0.035, 0.036) | 0.993 (0.954, 1.034) | 0.733 | 0.000 (−0.002, 0.001) | 0.732 | – |
| CP prediction or U09.9 diagnosis (90–365 days) | 0.088 (0.086, 0.090) | 0.088 (0.086, 0.089) | 1.002 (0.975, 1.031) | 0.866 | 0.000 (−0.002, 0.003) | 0.866 | – |
| No censoring at last visit date | 0.043 (0.042, 0.045) | 0.043 (0.042, 0.043) | 1.020 (0.987, 1.053) | 0.241 | 0.001 (−0.001, 0.002) | 0.250 | – |
| Exclude patients with other treatments | 0.045 (0.043, 0.046) | 0.045 (0.044, 0.046) | 0.992 (0.957, 1.029) | 0.660 | 0.000 (−0.002, 0.001) | 0.660 | – |
| Death as censoring event | 0.046 (0.044, 0.047) | 0.047 (0.046, 0.047) | 0.982 (0.948, 1.018) | 0.324 | −0.001 (−0.002, 0.001) | 0.322 | – |
| Doubly Robust Adjustment | Hazard Ratio: 0.983 (0.950, 1.018) | | | p = 0.305 | | | |

Abbreviations: CCI, Charlson comorbidity index; CI, confidence interval; CP, computable phenotype; PASC, post-acute sequelae of COVID-19; NNT, number needed to treat; VA, United States Department of Veterans Affairs.

In the CCI-stratified analysis, Paxlovid had a significant protective effect only among patients with a CCI score of 3–4 (RR = 0.83; ARD = −1.30%; NNT = 76). Paxlovid's effect among patients with a CCI score of 5–10 was similar in magnitude but not statistically significant (RR = 0.91; ARD = −0.80%). Among patients with a CCI of 11 or greater, the confidence intervals were wide, and the effect estimate was not statistically significant, limiting meaningful interpretation.

In the vaccination-stratified analysis, we found that Paxlovid's effect did not vary between unvaccinated and fully vaccinated patients.

Together, these stratified analyses suggest that Paxlovid may be more effective at preventing PASC among patients at higher-risk of PASC. However, we do not emphasize this interpretation for two reasons. First, effects are not monotonic across age or CCI strata, so that dynamic is plausible but not obvious. Second, even the largest effect sizes are small, with NNTs in the hundreds. Practically, a risk-stratified approach to Paxlovid as a PASC preventive is unlikely to lead to significant population-level changes in PASC incidence.

In the VA-like subanalysis, we also found no significant effects of Paxlovid on PASC overall. Two of our symptom cluster outcomes share many ICD-10 codes with PASC components measured in Xie and colleagues [14]. First, our respiratory symptom cluster (RR 0.92, unadjusted 0.94) aligns with their "shortness of breath" component (HR 0.89, unadjusted RR 0.82). Second, our fatigue symptom cluster (RR 0.93, unadjusted 0.91) aligns with their "fatigue and malaise" component (HR 0.79, unadjusted RR 0.70). This comparison suggests that cohort differences explain much of the difference between our findings and those of Xie and colleagues [14]. Our unadjusted RRs are very different for nearly identical PASC components, which suggests that different statistical methods do not explain the difference in findings. Other aspects of a true VA cohort may explain the remaining difference. Veterans are more likely than demographically similar

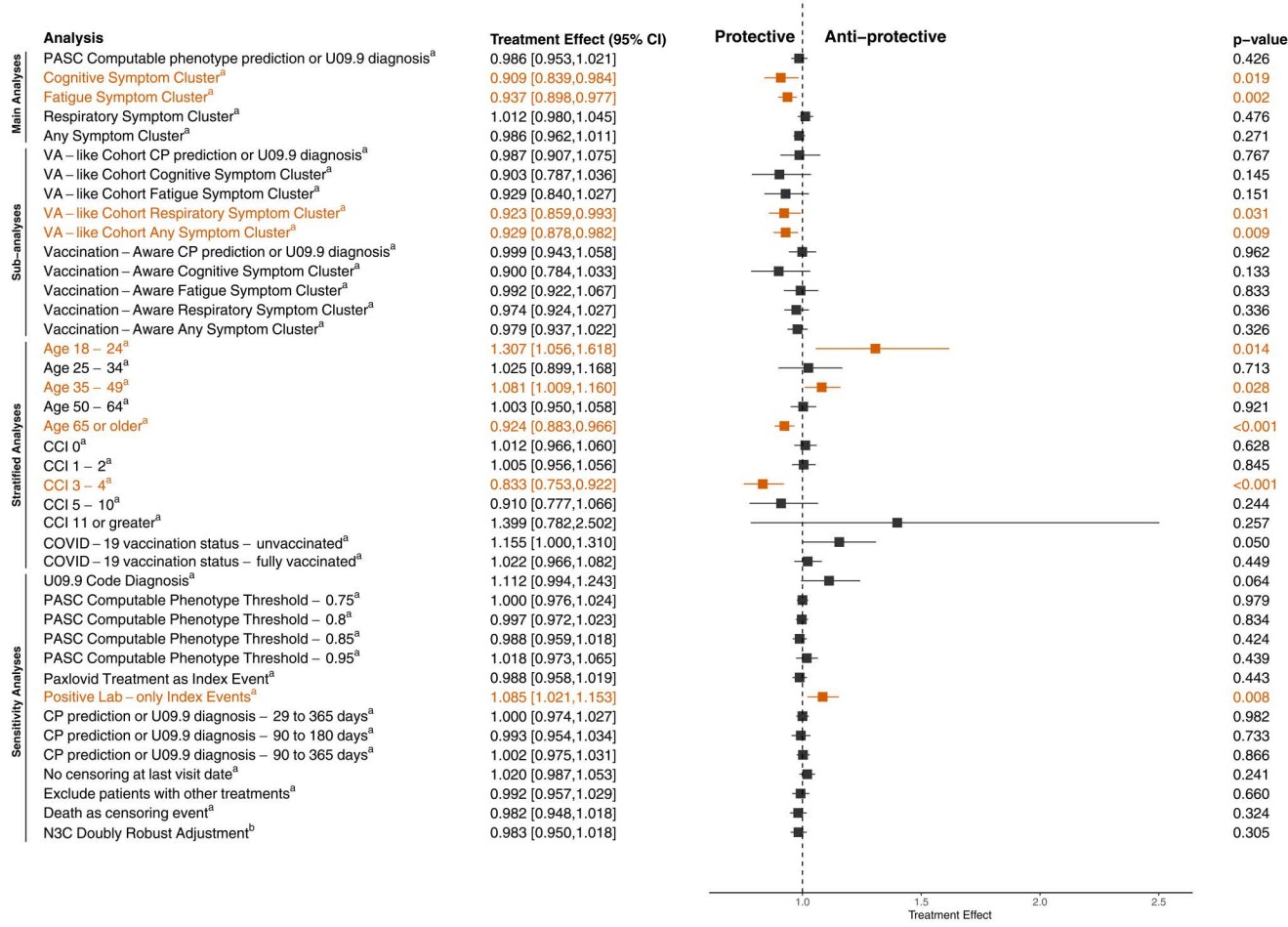

**Note:** [a]Analysis reports Relative Risk; [b]Analysis reports adjusted Hazard Ratio

**Fig 3. Estimated Treatment effects (risk ratios) of Paxlovid on PASC, across all analyses.** PASC, post-acute sequelae of COVID-19; VA, United States Department of Veterans Affairs; CCI, Charlson comorbidity index; CP, computable phenotype.

non-veterans to have been exposed to traumatic brain injury, post-traumatic stress disorder, biohazards, and other risk factors [39–43]. Through consistent access to the VA, the EHR for veterans may also be more complete [44]. Veterans may also differ from demographically similar non-veterans in their access to care.

In the vaccination-aware subanalysis, we also found no significant effects of Paxlovid on PASC. We interpret this to imply that vaccination did not cause unmeasured confounding in the primary analysis.

This study has several strengths that underscore the value of large-scale EHR repositories. We used a large, nationally sampled cohort from 28 sites across the United States, increasing generalizability and decreasing the potential for misclassification present in administrative or claims data [45]. The volume of data in the N3C database allowed for the aggressive inclusion/exclusion criteria necessary for TTE while preserving statistical power [46,47]. Our use of the TTE framework with inverse probability of censoring and treatment weighting allowed us to account for confounding and informative censoring and to estimate the causal effect of Paxlovid treatment using observational data [48–51]. Our use of a PASC computable phenotype is also a strength, as described in the Methods section. Finally, the study period makes our findings more relevant than prior studies of this topic, which have included cases from the initial Omicron wave, when Paxlovid was less available and disease dynamics were markedly different.

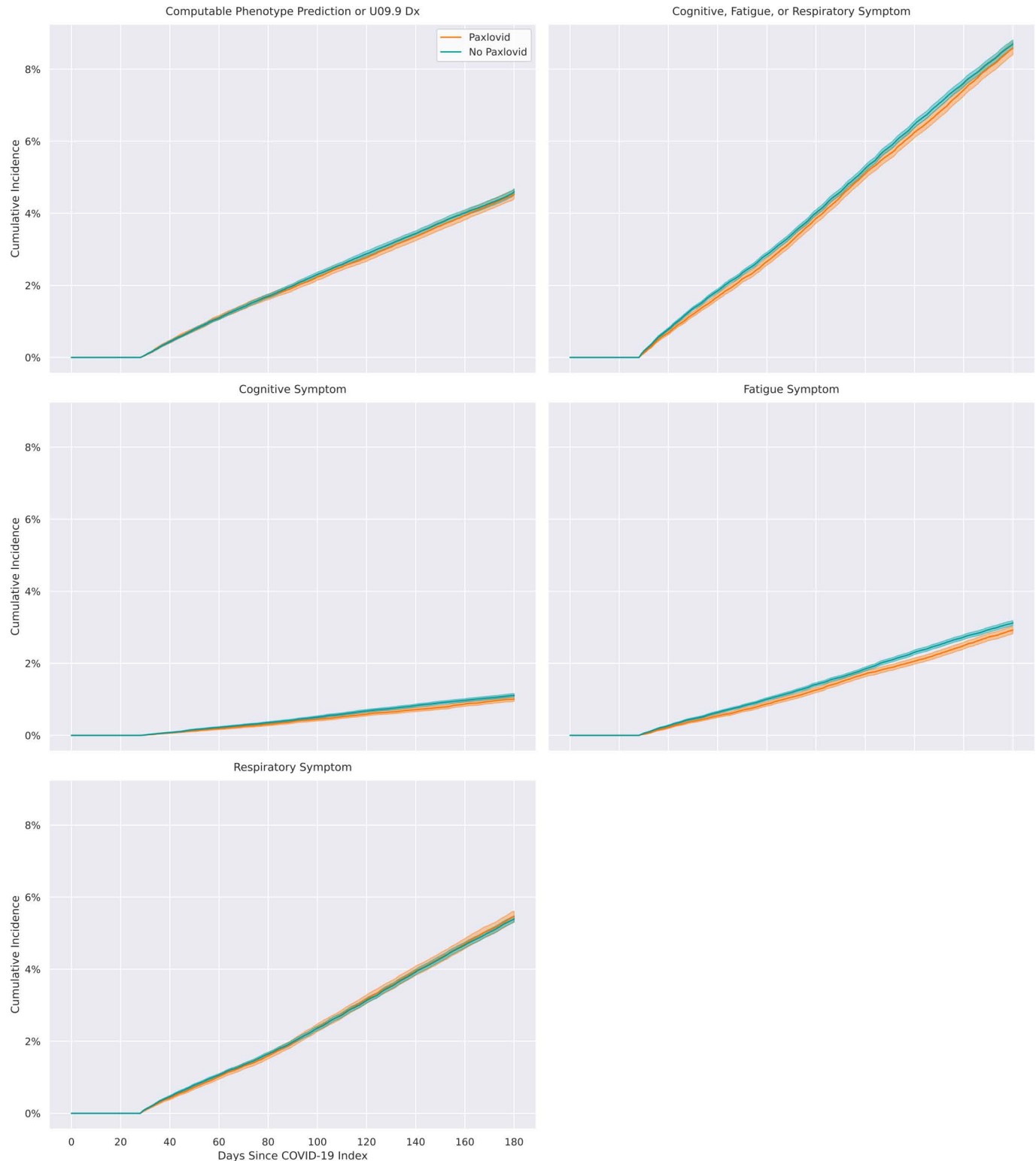

**Fig 4. Cumulative incidence of PASC in Paxlovid-treated vs. non-Paxlovid-treated patients by outcome measure, between 29 and 180 days.**

Three groups of limitations affected this study: (1) limitations related to the cohort definition; (2) limitations related to measurement; and (3) limitations related to observational studies.

This study's eligibility criteria included eligibility for on-label Paxlovid treatment (i.e., at risk for developing severe COVID-19 due to the presence of one or more risk factors). Therefore, results can only be generalized to a high-risk population. Ideally, a clinical trial of Paxlovid as a PASC preventive would also assess treatment among lower-risk populations. We chose not to emulate such a trial because it would complicate the study design and make exchangeability harder to establish due to confounding by indication. We note that the CanTreatCOVID trial also includes high-risk patients only. The effect of Paxlovid treatment on PASC onset among lower-risk patients is an area for future research. Additionally, this study's inclusion criterion of Paxlovid treatment within 5 days of COVID-19 index differed from the indication of treatment within 5 days of symptom onset. However, we note that within our cohort, 97% of patients in the treatment group were prescribed Paxlovid within 1 day of COVID-19 index.

Several variables in this study were subject to measurement error. Many COVID-19 cases during this period were not documented due to the prevalence of home testing, and patients with documented COVID-19 may not be representative of all patients with COVID-19. COVID-19 vaccination is often undocumented in the EHR, because vaccination at pharmacies and other care sites is common. Our vaccination subanalysis addressed this issue by focusing on sites with reliable vaccination data. Paxlovid prescriptions from providers outside N3C data partner systems may not be documented. The PASC computable phenotype may also misclassify patients [34]. For this reason, the confidence intervals around computable phenotype-based incidence estimates are likely too narrow. In addition to measurement error, different measures of PASC may account for some of the differences between studies on this topic. Although several institutions have proposed definitions of PASC, they disagree on the symptoms and timing that constitute the condition [52–55]. Finally, this study is subject to limitations common to EHR-based studies. EHRs are susceptible to missing data, and our estimates may be biased if missingness was informative [56–58].

This study is also subject to the assumptions of all causal inference studies, in particular, that there is no unmeasured confounding. One potential unmeasured confounder is acute COVID-19 severity prior to index. Sicker patients may be more likely to seek Paxlovid and develop PASC. The EHR contains no reliable measure of this construct, but we control for pre-diagnosis comorbidities, which have been shown to correlate so strongly with COVID-19 severity that they can be considered proxies, thus mitigating the potential unmeasured confounding from this source [59–61]. Propensity to seek healthcare and access to care may be additional unmeasured confounders, but we control for utilization in the prior year as a proxy for these constructs.

In conclusion, we see Paxlovid as unlikely to become a definitive solution for PASC prevention. Although it may have a small protective effect among higher-risk patients and on certain symptoms, the effect sizes are negligible. For example, an absolute risk reduction of 0.43% among patients aged 65 years or more means that 233 people would need to be treated with Paxlovid to prevent one case of PASC. Paxlovid remains an important tool to reduce the pandemic's public health burden by preventing hospitalization and death due to acute COVID-19. However, broadly effective interventions to prevent PASC remain elusive.

## Supporting information

**S1 Text. Supporting Information. Fig A:** Outcome co-occurrence matrix. Each cell represents the percentage of patients with the row outcome who also had the column outcome. **Fig B:** Cumulative incidence of PASC in Paxlovid-treated vs. non-Paxlovid-treated patients by outcome measure; between 29 and 180 days; VA-like subanalysis. **Fig C:** Cumulative incidence of PASC in Paxlovid-treated vs. non-Paxlovid-treated patients by predicted outcome from CP model with threshold of 0.9 or U09.9, additionally adjusted for vaccination status and among data partners meeting vaccination data quality criteria. **Fig D:** Causal diagram used to inform covariate selection. Treatment is shown in green; outcome is shown in orange; observed covariates are shown in gray; unobserved covariates are shown in pink. Note that this diagram

only shows the relationships relevant to this study. Covariates may have other causes that are omitted for clarity. **Table A:** ICD-10 codes used to define Global Burden of Disease symptom clusters.
(DOCX)

**S1 STROBE Checklist. STROBE Checklist.** The checklist is licensed under Creative Commons Attribution 4.0 International (CC BY 4.0) license.
(DOCX)

## Acknowledgments

This study is part of the NIH Researching COVID to Enhance Recovery (RECOVER) Initiative, which seeks to understand, treat, and prevent the post-acute sequelae of COVID-19 infection (PASC). For more information on RECOVER, visit https://recovercovid.org/.

The analyses described in this manuscript were conducted with data or tools accessed through the NCATS N3C Data Enclave https://covid.cd2h.org and N3C Attribution and Publication Policy v 1.2-2020-08-25b supported by NCATS U24 TR002306, Axle Informatics Subcontract: NCATS-P00438-B, and by the RECOVER Initiative (OT2HL161847–01). The N3C Publication committee confirmed that this manuscript msid: 1733.497 is in accordance with N3C data use and attribution policies; however, this content is solely the responsibility of the authors and does not necessarily represent the official views of the National Institutes of Health or the RECOVER or N3C programs. This research was possible because of the patients whose information is included within the data and the organizations (https://ncats.nih.gov/n3c/resources/data-contribution/data-transfer-agreement-signatories) and scientists who have contributed to the on-going development of this community resource (https://doi.org/10.1093/jamia/ocaa196). We would also like to thank the National Community Engagement Group (NCEG), all patients, caregivers, and community Representatives, and all the participants enrolled in the RECOVER Initiative.

We also acknowledge the following institutions whose data is released or pending:

Available: Advocate Health Care Network—UL1TR002389: The Institute for Translational Medicine (ITM) • Aurora Health Care Inc—UL1TR002373: Wisconsin Network For Health Research • Boston University Medical Campus—UL1TR001430: Boston University Clinical and Translational Science Institute • Brown University—U54GM115677: Advance Clinical Translational Research (Advance-CTR) • Carilion Clinic—UL1TR003015: iTHRIV Integrated Translational health Research Institute of Virginia • Case Western Reserve University—UL1TR002548: The Clinical & Translational Science Collaborative of Cleveland (CTSC) • Charleston Area Medical Center—U54GM104942: West Virginia Clinical and Translational Science Institute (WVCTSI) • Children's Hospital Colorado—UL1TR002535: Colorado Clinical and Translational Sciences Institute • Columbia University Irving Medical Center—UL1TR001873: Irving Institute for Clinical and Translational Research • Dartmouth College—None (Voluntary) Duke University—UL1TR002553: Duke Clinical and Translational Science Institute • George Washington Children's Research Institute—UL1TR001876: Clinical and Translational Science Institute at Children's National (CTSA-CN) • George Washington University—UL1TR001876: Clinical and Translational Science Institute at Children's National (CTSA-CN) • Harvard Medical School—UL1TR002541: Harvard Catalyst • Indiana University School of Medicine—UL1TR002529: Indiana Clinical and Translational Science Institute • Johns Hopkins University—UL1TR003098: Johns Hopkins Institute for Clinical and Translational Research • Louisiana Public Health Institute—None (Voluntary) • Loyola Medicine—Loyola University Medical Center • Loyola University Medical Center—UL1TR002389: The Institute for Translational Medicine (ITM) • Maine Medical Center—U54GM115516: Northern New England Clinical & Translational Research (NNE-CTR) Network • Mary Hitchcock Memorial Hospital & Dartmouth Hitchcock Clinic—None (Voluntary) • Massachusetts General Brigham—UL1TR002541: Harvard Catalyst • Mayo Clinic Rochester—UL1TR002377: Mayo Clinic Center for Clinical and Translational Science (CCaTS) • Medical University of South Carolina—UL1TR001450: South Carolina Clinical & Translational Research Institute (SCTR) •

MITRE Corporation—None (Voluntary) • Montefiore Medical Center—UL1TR002556: Institute for Clinical and Translational Research at Einstein and Montefiore • Nemours—U54GM104941: Delaware CTR ACCEL Program • NorthShore University HealthSystem—UL1TR002389: The Institute for Translational Medicine (ITM) • Northwestern University at Chicago—UL1TR001422: Northwestern University Clinical and Translational Science Institute (NUCATS) • OCHIN—INV-018455: Bill and Melinda Gates Foundation grant to Sage Bionetworks • Oregon Health & Science University—UL1TR002369: Oregon Clinical and Translational Research Institute • Penn State Health Milton S. Hershey Medical Center—UL1TR002014: Penn State Clinical and Translational Science Institute • Rush University Medical Center—UL1TR002389: The Institute for Translational Medicine (ITM) • Rutgers, The State University of New Jersey—UL1TR003017: New Jersey Alliance for Clinical and Translational Science • Stony Brook University—U24TR002306 • The Alliance at the University of Puerto Rico, Medical Sciences Campus—U54GM133807: Hispanic Alliance for Clinical and Translational Research (The Alliance) • The Ohio State University—UL1TR002733: Center for Clinical and Translational Science • The State University of New York at Buffalo—UL1TR001412: Clinical and Translational Science Institute • The University of Chicago—UL1TR002389: The Institute for Translational Medicine (ITM) • The University of Iowa—UL1TR002537: Institute for Clinical and Translational Science • The University of Miami Leonard M. Miller School of Medicine— UL1TR002736: University of Miami Clinical and Translational Science Institute • The University of Michigan at Ann Arbor—UL1TR002240: Michigan Institute for Clinical and Health Research • The University of Texas Health Science Center at Houston—UL1TR003167: Center for Clinical and Translational Sciences (CCTS) • The University of Texas Medical Branch at Galveston—UL1TR001439: The Institute for Translational Sciences • The University of Utah—UL1TR002538: Uhealth Center for Clinical and Translational Science • Tufts Medical Center—UL1TR002544: Tufts Clinical and Translational Science Institute • Tulane University—UL1TR003096: Center for Clinical and Translational Science • The Queens Medical Center—None (Voluntary) • University Mdical Center New Orleans—U54GM104940: Louisiana Clinical and Translational Science (LA CaTS) Center • University of Alabama at Birmingham—UL1TR003096: Center for Clinical and Translational Science • University of Arkansas for Medical Sciences—UL1TR003107: UAMS Translational Research Institute • University of Cincinnati—UL1TR001425: Center for Clinical and Translational Science and Training • University of Colorado Denver, Anschutz Medical Campus—UL1TR002535: Colorado Clinical and Translational Sciences Institute • University of Illinois at Chicago—UL1TR002003: UIC Center for Clinical and Translational Science • University of Kansas Medical Center—UL1TR002366: Frontiers: University of Kansas Clinical and Translational Science Institute • University of Kentucky—UL1TR001998: UK Center for Clinical and Translational Science • University of Massachusetts Medical School Worcester—UL1TR001453: The UMass Center for Clinical and Translational Science (UMCCTS) • University Medical Center of Southern Nevada—None (voluntary) • University of Minnesota—UL1TR002494: Clinical and Translational Science Institute • University of Mississippi Medical Center—U54GM115428: Mississippi Center for Clinical and Translational Research (CCTR) • University of Nebraska Medical Center—U54GM115458: Great Plains IDeA-Clinical & Translational Research • University of North Carolina at Chapel Hill—UL1TR002489: North Carolina Translational and Clinical Science Institute • University of Oklahoma Health Sciences Center—U54GM104938: Oklahoma Clinical and Translational Science Institute (OCTSI) • University of Pittsburgh—UL1TR001857: The Clinical and Translational Science Institute (CTSI) • University of Pennsylvania—UL1TR001878: Institute for Translational Medicine and Therapeutics • University of Rochester—UL1TR002001: UR Clinical & Translational Science Institute • University of Southern California—UL1TR001855: The Southern California Clinical and Translational Science Institute (SC CTSI) • University of Vermont—U54GM115516: Northern New England Clinical & Translational Research (NNE-CTR) Network • University of Virginia—UL1TR003015: iTHRIV Integrated Translational health Research Institute of Virginia • University of Washington—UL1TR002319: Institute of Translational Health Sciences • University of Wisconsin-Madison—UL1TR002373: UW Institute for Clinical and Translational Research • Vanderbilt University Medical Center—UL1TR002243: Vanderbilt Institute for Clinical and Translational Research • Virginia Commonwealth University—UL1TR002649: C. Kenneth and Dianne Wright Center for Clinical and Translational Research • Wake Forest University Health Sciences—UL1TR001420: Wake

Forest Clinical and Translational Science Institute • Washington University in St. Louis—UL1TR002345: Institute of Clinical and Translational Sciences • Weill Medical College of Cornell University—UL1TR002384: Weill Cornell Medicine Clinical and Translational Science Center • West Virginia University—U54GM104942: West Virginia Clinical and Translational Science Institute (WVCTSI) Submitted: Icahn School of Medicine at Mount Sinai—UL1TR001433: ConduITS Institute for Translational Sciences • The University of Texas Health Science Center at Tyler—UL1TR003167: Center for Clinical and Translational Sciences (CCTS) • University of California, Davis—UL1TR001860: UCDavis Health Clinical and Translational Science Center • University of California, Irvine—UL1TR001414: The UC Irvine Institute for Clinical and Translational Science (ICTS) • University of California, Los Angeles—UL1TR001881: UCLA Clinical Translational Science Institute • University of California, San Diego—UL1TR001442: Altman Clinical and Translational Research Institute • University of California, San Francisco—UL1TR001872: UCSF Clinical and Translational Science Institute Pending: Arkansas Children's Hospital—UL1TR003107: UAMS Translational Research Institute • Baylor College of Medicine— None (Voluntary) • Children's Hospital of Philadelphia—UL1TR001878: Institute for Translational Medicine and Therapeutics • Cincinnati Children's Hospital Medical Center—UL1TR001425: Center for Clinical and Translational Science and Training • Emory University—UL1TR002378: Georgia Clinical and Translational Science Alliance • HonorHealth—None (Voluntary) • Loyola University Chicago—UL1TR002389: The Institute for Translational Medicine (ITM) • Medical College of Wisconsin—UL1TR001436: Clinical and Translational Science Institute of Southeast Wisconsin • MedStar Health Research Institute—None (Voluntary) • Georgetown University—UL1TR001409: The Georgetown-Howard Universities Center for Clinical and Translational Science (GHUCCTS) • MetroHealth—None (Voluntary) • Montana State University— U54GM115371: American Indian/Alaska Native CTR • NYU Langone Medical Center—UL1TR001445: Langone Health's Clinical and Translational Science Institute • Ochsner Medical Center—U54GM104940: Louisiana Clinical and Translational Science (LA CaTS) Center • Regenstrief Institute—UL1TR002529: Indiana Clinical and Translational Science Institute • Sanford Research—None (Voluntary) • Stanford University—UL1TR003142: Spectrum: The Stanford Center for Clinical and Translational Research and Education • The Rockefeller University—UL1TR001866: Center for Clinical and Translational Science • The Scripps Research Institute—UL1TR002550: Scripps Research Translational Institute • University of Florida—UL1TR001427: UF Clinical and Translational Science Institute • University of New Mexico Health Sciences Center—UL1TR001449: University of New Mexico Clinical and Translational Science Center • University of Texas Health Science Center at San Antonio—UL1TR002645: Institute for Integration of Medicine and Science • Yale New Haven Hospital—UL1TR001863: Yale Center for Clinical Investigation

## Author contributions

**Conceptualization:** Alexander Preiss, Abhishek Bhatia, Leyna V. Aragon, John M. Baratta, Frank Blancero, Michael Daniel Brannock, Robert F. Chew, Iván Díaz, Megan Fitzgerald, Thomas W. Carton, Christopher G. Chute, Melissa Haendel, Richard Moffitt, Emily Pfaff.

**Data curation:** Alexander Preiss, Abhishek Bhatia, Michael Daniel Brannock, Robert F. Chew, Andrea G. Zhou, Richard Moffitt, Emily Pfaff.

**Formal analysis:** Alexander Preiss, Abhishek Bhatia, Monika Baskaran, Michael Daniel Brannock, Iván Díaz.

**Funding acquisition:** Thomas W. Carton, Christopher G. Chute, Melissa Haendel, Richard Moffitt, Emily Pfaff.

**Investigation:** Alexander Preiss, Abhishek Bhatia, Leyna V. Aragon, John M. Baratta, Frank Blancero, Iván Díaz, Megan Fitzgerald, Christopher G. Chute, Melissa Haendel, Richard Moffitt, Emily Pfaff.

**Methodology:** Alexander Preiss, Abhishek Bhatia, Michael Daniel Brannock, Robert F. Chew, Iván Díaz, Megan Fitzgerald, Thomas W. Carton, Richard Moffitt, Emily Pfaff.

**Project administration:** Alexander Preiss, Elizabeth P. Kelly.

**Software:** Alexander Preiss, Abhishek Bhatia, Michael Daniel Brannock, Andrea G. Zhou.

**Supervision:** Alexander Preiss, Elizabeth P. Kelly, Emily Pfaff.

**Validation:** Alexander Preiss, Abhishek Bhatia, John M. Baratta.

**Visualization:** Alexander Preiss, Abhishek Bhatia, Monika Baskaran.

**Writing – original draft:** Alexander Preiss, Abhishek Bhatia, Leyna V. Aragon, Michael Daniel Brannock, Emily Pfaff.

**Writing – review & editing:** Alexander Preiss, Abhishek Bhatia, Leyna V. Aragon, John M. Baratta, Monika Baskaran, Frank Blancero, Michael Daniel Brannock, Robert F. Chew, Iván Díaz, Megan Fitzgerald, Elizabeth P. Kelly, Andrea G. Zhou, Thomas W. Carton, Christopher G. Chute, Melissa Haendel, Richard Moffitt, Emily Pfaff.

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
