## [Editor Report · Decision Letter 0]

30 Jul 2024

Dear Dr Preiss,

Thank you for submitting your manuscript entitled "Effect of Paxlovid Treatment during Acute COVID-19 on Long COVID Onset: An EHR-Based Target Trial Emulation from the N3C and RECOVER Consortia" for consideration by PLOS Medicine.

Your manuscript has now been evaluated by the PLOS Medicine editorial staff and I am writing to let you know that we would like to send your submission out for external peer review.

We note your related manuscript (24-00548) currently under consideration at PLOS Medicine and your request to (potentially) co-publish with this manuscript. This is certainly something we could consider further down the line but we really do need to see the revised version of your existing manuscript and also see how this one fares during the peer review process before making any decisions regarding publication of either. It would have been helpful to us if the revised version of 24-00548 had been returned earlier to help inform our decision making and we are very keen that you resubmit the former as soon as possible.

Before we can send your manuscript to reviewers, we need you to complete your submission by providing the metadata that is required for full assessment. To this end, please login to Editorial Manager where you will find the paper in the 'Submissions Needing Revisions' folder on your homepage. Please click 'Revise Submission' from the Action Links and complete all additional questions in the submission questionnaire.

Please re-submit your manuscript within two working days, i.e. by Aug 01 2024 11:59PM.

Feel free to email me at pdodd@plos.org or the team at plosmedicine@plos.org if you have any queries relating to your submission.

Kind regards,

Pippa

Philippa Dodd, MBBS MRCP PhD

Senior Editor

PLOS Medicine

pdodd@plos.org

---

## [Decision Letter · Decision Letter 1]

23 Sep 2024

Dear Dr Preiss,

Many thanks for submitting your manuscript "Effect of Paxlovid Treatment during Acute COVID-19 on Long COVID Onset: An EHR-Based Target Trial Emulation from the N3C and RECOVER Consortia" (PMEDICINE-D-24-02349R1) to PLOS Medicine. The paper has been reviewed by subject experts and a statistician; their comments are included below and can also be accessed here: [LINK]

As you will see, the reviewers were very positive about the paper but, they raised a number of questions about specific study details and the methodological approach. After discussing the paper with the editorial team and an academic editor with relevant expertise, I'm pleased to invite you to revise the paper in response to the comments detailed below. We plan to send the revised paper to some or all of the original reviewers, and we cannot provide any guarantees at this stage regarding publication.

We ask that you submit your revision by Oct 14 2024 11:59PM. However, if this deadline is not feasible, please contact me by email, and we can discuss a suitable alternative.

Don't hesitate to contact me directly with any questions (pdodd@plos.org).

Best regards,

Pippa

Philippa Dodd, MBBS MRCP PhD

Senior Editor

PLOS Medicine

pdodd@plos.org

Comments from the academic editor:

I think a major revision is appropriate.

1) It does seem that the authors used the TTE and cloning-censoring-weighting approach as they did with the last manuscript. However, the aspect of cloning and censoring is not well described.

2) I think there is potential for bias to exclude individuals who were hospitalized prior to receipt of Paxlovid from the treatment arm. In a hypothetical RCT where treatment is assigned on the date of diagnosis, individuals who are hospitalized prior to treatment with paxlovid would still be considered in the treatment arm (if the outcome is PASC). In this case, being hospitalized becomes a competing risk a precludes some individuals from being in treatment arm but not control.

3) Death is certainly a competing risk but authors still chose to censor. I was not clear of the explanation on how competing risk is considered a mediator. In this circumstance, it could be particularly relevant to treat appropriately as competing risk. Theoretically, Paxlovid could reduce mortality, but those in whom death was prevented could be at much higher risk of PASC. In that sense, Paxlovid could essentially be converting deaths into alive, but PASC. Understanding that full picture with the appropriate use of competing risk seems relevant to understand incidence of both. Fine gray models could be used for measuring HR.

4) I think more discussion on the importance and relevance of using the phenotypic model for the outcome is important. Initially, I wanted to see a sensitivity analysis where they only used documented ICD 10 codes as the outcome. But then I saw that in the results, which interesting showed more PASC with treatment. In hindsight, this might make sense as people with more healthcare access are likely to get Paxlovid and evaluated for PASC. I think a discuss of the pros adn cons is important. Initially, I was wondering whether it added additional complexity and opaqueness without a clear benefit. This issue should be addressed clearly (it may be in the cited papers but some discussion needs to be in methods as well).

5) In comparison with Xie et al, would also want some comments on similarities/differences in the approaches taken. It states in discussion that methodology is similar but I couldn't see much beyond that. If authors analyzed the VA dataset using the same method they use here, would they anticipate getting same results as Xie et al or their own? Based on discussion, they seem to be assuming that they would get same results as Xie et al as they attribute differences mostly to the population.

6) Stratified estimates could also be of potential interest. Particular based on age, vaccination status( i.e., effect in vaccinated and not vaccinated population, not adjusting for vaccination).

Comments from the reviewers:

Reviewer #1: Thank you for the opportunity to review this manuscript. Overall, the authors made a very good attempt to answer the question at hand, but there some pertinent points that need addressing.

1. The manuscript is too long. As an insight, the introduction is nearly 2 pages and discussion is nearly 4 pages. There is also a lot of repetition. To make it easier to read, considerable effort should be made to write in a concise manner. A lot of the detail could be moved to an appendix.

2. The treatment strategies of interest are unclear. In Table 3, the treatment strategy for only 1 arm is outlined: the paxlovid arm. Even within this description, there is no information on how long treatment should be taken for, or if patients are allowed to stop for any reason. There should then additionally be detailed information on the treatment strategy for the "no treatment" arm. Should patients continue not to take treatment for the duration of follow up? Reading the text, it seems this is the case as individuals that are not prescribed at baseline are censored if the start treatment at a later date. However, all this information should be made clear in the definition of the treatment strategies.

3. Follow on from above, the causal contrast row in Table 3 states that the investigators aim to estimate the intention-to-treat effect. This is the effect of assignment to one of the treatment arms at baseline, regardless of what happens later. However, in the "no treatment" group, they censor when someone starts treatment. This, therefore, is not akin to an ITT effect. If the investigators do censor at initiation in the no treatment arm, then they should also censor at non-adherence to the treatment strategy in the treatment arm. This would then be more similar to a per protocol effect and would also requite adjusting for baseline and time-updated predictors of non-adherence. But, this also requires stricter definition of what the treatment strategies are, as specified above.

4. There is a misalignment between assignment and when outcomes start to be collected. Treatment with paxlovid is assessed within 5 days after COVID diagnosis, but individuals are also censored if they have an event within the first 28 days after COVID. So, outcomes only start to be counted after 28 days. Why is this? It leave the study prone to immortal time.

5. Follow up can end at the time of the last recorded visit? Please can the investigators expand on what this means?

6. As paxlovid is known to reduce the risk of hospitalization, hospitalization could be a "semi-competing event". This is because people that are hospitalized may receive other care and treatments that in turn reduce their risk of PASC. If those with paxlovid are less likely to be hospitalized, then the effect estimated in this study is not interpretable. Please can the investigators clarify how hospitalization is dealt with, and how they have handled the possibility this this is a competing event.

6. How was missing data handled?

7. Please run the analyses with at least 500 bootstrap samples.

8. The discussion states that using the target trial framework allowed to account for confounding. This is not what the target trial framework does. It is a framework to formalize study design, but the only way to account for confounding it to adjust analyses for baseline variables.

Reviewer #2: Thank you for this opportunity to review this article reporting a study emulating a target trial on Paxlovid in association with a hypothesized lower risk of PASC, i.e., long Covid.

I find the writing to be clear and organized, with results presented comprehensively. There are the following points I would like to raise for the authors' consideration:

1. The index date was set at Covid diagnosis or positive SARS-CoV-2 test. What if the person gets Covid and was about to be prescribed Paxlovid within 5 days but didn't because they were admitted to the ICU or unconscious? What if the person dies 2 days of Covid diagnosis and did not make it to receive Paxlovid prescription? Would these individuals cause bias to the results and how did the authors address this? Why did the authors' not use cloning-censoring-weighting, for which Miguel Hernan is highly supportive, to address this challenge? Also see this example for your reference: https://doi.org/10.7326/M22-3057

2. Maybe I missed it but I did not see methods describing the approach to address other antivirals, immunotherapy or other pharmacotherapies. Hence, the non-use group is a diverse group of people, possibly using different therapies. For example, remdesivir was quite commonly used for Covid for a while. What about molunupiravir?

3. Hazard ratio is roughly an average of the ratios of hazard rates between the exposed and unexposed groups across the observation period, but only considering the surviving cohort at each time point. In a clinical trial, which this study aims to emulate, hazard ratios are not commonly used because randomization is only useful if the comparison is between the groups in full.

Reviewer #3: I think this is a high-quality analysis with detailed and careful reporting that will be of interest to the journal readers who may be interested in the methodology or the generated evidence. I have a few major comments as summarized below.

Major comments

1- It would be great if not essential to replicate the results of an RCT first before moving on to estimate effects that are not yet evaluated by trials. The authors have mentioned a few of those possible trials in the Introduction. I suppose it is also possible to replicate the effect of Paxlovid on mortality. Another option would be to choose a negative control outcome.

2- If #1 is doable, I would replace the VA analysis with this new evidence as replicating another observational analyses with a new one is not as informative. As the authors have correctly mentioned, there is always the possibility of effect measure modification in addition to confounding which makes the results much less interesting.

3- The estimated effects on cognitive function and fatigue are rather small. The authors have labeled similar effects in other analyses as practically insignificant. I suggest using the same qualification for these effects in the Abstract, Results and Discussion, especially considering that as the authors have mentioned, these are rather uncommon outcomes.

4- I would suggest revising the causal graph by removing the box from the nodes that are not directly adjusted for in the analyses. That will allow the readers to identify potential bias pathways.

Minor comments

1- There are too many tables and figures. I suggest moving Figures 4, and 7-10 as well as Tables 1 and 5 to the online Appendix.

2- Please clarify if the protocol was registered before conducting the analyses.

3- Page 12, much of the second paragraph can be moved to the Methods.

Reviewer #4:

I have a number of comments that I hope the Editor and Authors will be find helpful:

MAJOR COMMENTS

Abstract: control treatment is not defined or mentioned

Outcome

-Symptom clusters were defined based on 'novel onset of PASC symptoms'. Please specify if only individuals with PASC based on the algorithm were assessed for symptoms-specific outcomes. Please specify what 'novel' means and if the clusters are mutually exclusive or if the same individuals could contribute to more than one symptom cluster.

-I find it difficult to understand why individuals hospitalized within 5 days of COVID before treatment were censored while individuals who were hospitalized within 5 days of COVID and had no treatment were not censored. This can introduce bias.

-Given that Paxlovid is indicated for individuals with moderate or severe COVID-19 symptoms, it is unclear why individuals who were hospitalized on the day of diagnosis were excluded.

-A treatment window/grace period of 5 days was used based on guidelines and this seems a reasonable choice. However, in real-world clinical practice, Paxlovid treatment might have been delayed beyond the 5-day time window in some individuals who then be classified as controls. Please present the distribution of time (median [IQR]) between COVID-19 index date and treatment initiation to show that the majority of Paxlovid treatments occurred within the 5-day window.

Weights

-Treatment assignment was defined over a 5-day window or grace period. In statistical analysis, please expand on how the weights were estimated to account for the time-varying nature of the treatment assignment. Please specify if a logistic regression model or another approach was used to estimate the probability of treatment. Please describe who this model was specified: was it a model for the daily probability of receiving Paxlovid or a model for the probability of receiving Paxlovid within 5 days of COVID index date?

-Please explain how the weights were stabilized. Usually stabilized weights imply two models, one for the denominator (to adjust for confounding) and one in the numerator (mostly to improve precision).

-My understanding is that the cumulative incidence curve was estimated by giving each individual a time-fixed weight equal (or proportional) to the inverse of the probability of their observed Paxlovid treatment status during the 5-day window. If so, please expand in the statistical analysis section using this or similar language.

Confounders

- Thank you for explaining the rationale for choosing the confounders! However, the variables mentioned in the text do not directly match the variables in the DAG. Consider updating the DAG.

- Please briefly report the distribution of the weights in results.

Follow-up and censoring

- When did follow-up start? At COVID-19 index date or 28 days after COVID-19 index date?

- I don't think follow-up should be censored at a PASC occurring within 28 days given that the outcome is defined as PASC after 28 days. These events should just be discarded and Individuals who have PASC symptoms during the time window should be allowed to experience PASC after 28 days.

- Because there is no adjustment for censoring, all results rely on the strong assumption that censoring is non-informative, which means that individuals who were censored are a random sample of all included individuals. This is usually not the case. I suggest rerunning the analyses using censoring weights. Each individual would receive weights equal to the inverse of the probability of being uncensored.

- Regarding death as a competing risk, thank you for clarifying that the focus is on the direct rather than the total effect. However, I recommend deleting the sentence 'The total effect would include any effect of Paxlovid treatment on PASC incidence that is mediated by death, which is less interpretable'. Technically this sentence is incorrect. Also, the direct effect is also problematic because 1) it quantified the effect of Paxlovid versus no Paxlovid in a world where patients cannot die and are immortal which is unrealistic and 2) it typically requires adjustment for selection bias. In the case of this article, I would say that choosing the direct or total effect does not matter much because mortality is relatively uncommon given the short follow-up.

- Thank you for trying a doubly-robust approach in Sensitivity Analyses. I would change or delete the sentence 'Targeted maximum likelihood estimation was not feasible with our cohort and computing environment…'. TMLE is only one of many doubly robust methods. You could say 'Computer intensive doubly robust methods like target maximum likelihood estimation were not feasible with our computing environment. ' Notice that the hazard ratio estimated from the Cox model using IPW and baseline adjustment is not directly comparable with a risk ratio because hazard and risks are mathematically different.

Target trial emulation (table 3)

-Control treatment is not defined

-Follow-up: text in the emulated trial is cut. Please specify the beginning and end of the follow-up. I don't see any mention here that follow-up was censored when PASC was recorded in the first 28 days. I see it is mentioned in Outcome, but it should be moved here. Here it is mentioned that individuals in the control group who received Paxlovid after day 5 are censored, but this was mentioned in the text!

MINOR COMMENTS

-I would avoid using the word 'significant' and 'unsignificant' throughout the text.

-Eligibility criteria (first paragraph): replace 'We excluded the period', with 'Individuals with COVID-19 diagnosis between December xxx '. Also, to improve clarify, I would move this sentence to the last paragraph of Eligibility criteria where all the exclusion criteria are listed.

-The operational definition of the outcome, PASC, is reported twice in overview and treatment and outcome sections. Please consolidate.

-Statistical analysis (second row): the author used the work 'rate'. Do they mean risk?

-Also, in the DAG, all variables seem to be 'descendants' of age and demographics only. Therefore, adjusting for age and demographics would be sufficient to control confounding…

-In statistical analysis, I recommend replacing 'We censored patients at the following events' with something like: 'Follow-up started at xx and ended at the earliest day of the outcome, death, last documented period, ect.'

---

* Please upload any figures associated with your paper as individual TIF or EPS files with 300dpi resolution at resubmission; please read our figure guidelines for more information on our requirements: http://journals.plos.org/plosmedicine/s/figures. While revising your submission, please upload your figure files to the PACE digital diagnostic tool, https://pacev2.apexcovantage.com/. PACE helps ensure that figures meet PLOS requirements. To use PACE, you must first register as a user. Then, login and navigate to the UPLOAD tab, where you will find detailed instructions on how to use the tool. If you encounter any issues or have any questions when using PACE, please email us at PLOSMedicine@plos.org.

FIGURES AND TABLES

SUPPLEMENTARY MATERIAL

In the published article, supporting information files are accessed only through a hyperlink attached to the captions. For this reason, you must list captions at the end of your manuscript file. You may include a caption within the supporting information file itself, as long as that caption is also provided in the manuscript file. Do not submit a separate caption file.

When supplementary files are contained with a single file:

Please label the file as ‘S1 Supporting Information’.

Please apply alphabetical labelling to each table and figure contained within the S1 file. For example, ‘Fig A’ to ‘Fig Z’ and ‘Table A’ to ‘Table Z’.

Plain text does not need to be labelled and can just be given a title as necessary. For example, ‘Statistical Analysis Plan’.

Please cite tables/figures as ‘Fig A in S1 Supporting Information’ and/or ‘Table A in S1 Supporting Information’, for example.

Please cite plain text as, ‘Statistical Analysis Plan in S1 Supporting Information’, for example.

When supplementary files are uploaded as separate files:

Please label tables as ‘S1 Table’ (so on)

Please label figures as ‘S1 Fig’ (and so on)

Any additional documents (protocols/analysis plans etc.) can be labelled as ‘S1 Protocol’, for example.

Please cite items as exactly as labelled.”

REFERENCES

OBSERVATIONAL STUDIES

* Abstract: Please include the study design, population and setting, number of participants, years during which the study took place (enrollment and follow up), length of follow up, and main outcome measures.

* MEDIATION ANALYSES - We recommend that the study is reported according to the AGReMA statement (https://agrema-statement.org/) and include the completed checklist as Supporting Information. Please add the following statement, or similar, to the Methods: "This study is reported as per the Guideline for Reporting Mediation Analyses (AGReMA) statement (S1 Checklist)." When completing the checklist, please use section and paragraph numbers, rather than page numbers.

* For all observational studies, in the manuscript text, please indicate: (1) the specific hypotheses you intended to test, (2) the analytical methods by which you planned to test them, (3) the analyses you actually performed, and (4) when reported analyses differ from those that were planned, transparent explanations for differences that affect the reliability of the study's results. If a reported analysis was performed based on an interesting but unanticipated pattern in the data, please be clear that the analysis was data driven.

* Please state in the Methods section whether the study had a prospective protocol or analysis plan. If a prospective analysis plan (from your funding proposal, IRB or other ethics committee submission, study protocol, or other planning document written before analyzing the data) was used in designing the study, please include the relevant document(s) with your revised manuscript as a Supporting Information file to be published alongside your study and cite it in the Methods section. A legend for this file should be included at the end of your manuscript. If no such document exists, please make sure that the Methods section transparently describes when analyses were planned, and when/why any data-driven changes to analyses took place. Changes in the analysis, including those made in response to peer review comments, should be identified as such in the Methods section of the paper, with rationale.

---

## [Decision Letter · Decision Letter 2]

10 Jan 2025

Dear Dr Preiss,

Many thanks for submitting your revised manuscript "Effect of Paxlovid Treatment during Acute COVID-19 on Long COVID Onset: An EHR-Based Target Trial Emulation from the N3C and RECOVER Consortia" (PMEDICINE-D-24-02349R2). The paper has been re-reviewed by two of the original subject experts and the statistical reviewer. Their comments are included below and can also be accessed here: [LINK]

As you will see, the subject reviewers were satisfied with the revision and your responses to their original comments; however, the statistician and academic editor raised some additional points of concern that we feel must be addressed in another revision. As such, I'm pleased to invite you to revise the paper in response to the reviewers' and editor’s comments. We may or may not undertake an additional round of reviews, and we cannot provide any guarantees at this stage regarding publication.

When you upload your revision, please once again include a point-by-point response that addresses all of the reviewer and editorial points, indicating the changes made in the manuscript and either an excerpt of the revised text or the location (eg: page and line number) where each change can be found. Please also be sure to check the general editorial comments at the end of this letter and include these in your point-by-point response. When you resubmit your paper, please include a clean version of the paper as the main article file and a version with changes tracked as a marked-up manuscript. It may also be helpful to check the guidelines for revised papers at http://journals.plos.org/plosmedicine/s/revising-your-manuscript for any that apply to your paper.

We ask that you submit your revision by Jan 31 2025 11:59PM. However, if this deadline is not feasible, please contact me by email, and we can discuss a suitable alternative. Please also contact me directly with any questions (hvanepps@plos.org).

Kind regards,

Heather

Heather Van Epps, PhD

Executive Editor

PLOS Medicine

hvanepps@plos.org

Comments from the academic editor:

The treatment strategies are still not well identified, and I think this has to do with the lack of a very discrete time zero and the use of a grace period. Currently, some post time-zero events are used to categorize individuals (e.g., if they are hospitalized in the first 5 days). Paxlovid in a non-hospitalized setting vs hospitalized setting is not a discrete distinction of treatment strategies (i.e., one would never randomize people to receive paxlovid in the hospital vs. not in the hospital…the selection bias in the different treatment arms would not make estimates very useful). Biologically, paxlovid works the same, but how sick the patient is, etc, may certainly moderate effects. In particular, “patients who received Paxlovid within 5 days, but during a hospitalization, were placed in the control group” is problematic. Hospitalization can be included in the overall eligibility criteria, but post time zero hospitalization can’t be used to categorize exposure. It really does not make any sense to define exposure as paxlovid treatment within 5 days but then categorize peopel who were treated in the first 5 days but hospitalized as control. I recognize “in outpatient setting” is included in the definition, but this has no relevance clinically. No one would ever hospitalize or not hospitalize someone in order to give paxlovid to follow this definition. If distinguishing the effect modification by hospitilization is of interest, the analysis has to be designed differently.

One way for the authors to address this is simply to have day 5 after treatment as time zero. Individuals who are not hospitalized are then categorized as to whether they have initiated paxlovid or not at that time point (individuals who are hospitalized within the first 5 days are excluded). However, this approach could create selection bias if there are differences in hospitalization that occur in those first 5 days. This would argue for the cloning-censoring approach, where treatment regimes can be very clearly specified and individuals can be under observation or censored based on what treatment regime is being assessed. The authors say there is no issue with immortal time bias, but the way they handled censoring certainly introduces it (i.e., to be exposed to intervention you need to stay out of hospital for at least 5 days and that is not the case in the control). I really do think the authors should consider this approach (even though findings may not change dramatically).

It is good to see that there is a supplemental analysis where deaths were treated as a competing risk and results were similar. I think this should be primary, which should not be difficult as the authors are already using the aalen-johansen approach. The question of interest is if 100 people are treated vs. 100 people are not, what happens to those 100 people. If there are differences in deaths, that is part of the story. Censoring deaths essentially changes the denominator in which you are assessing PASC. Treating it as a competing helps to maintain the denominator as 100. I would strongly argue this is the estimate of most interest, rather than trying to control for the aspect that is mediated by death.

A lot of this discussion on the epidemiologic methods and rationale is in the supplement but is not even referred to in the main text. A reader would not even know that they should check the supplement for relevant details.

Although stratified estimates may take some time, they are still relevant and important to do (and extensions to complete and comprehensive revision can be granted).

Ultimately, I do not disagree with the authors' conclusion (and several of the methodological challenges would likely bias towards favoring paxlovid), but there have been a lot of COVID studies and findings based on less than rigorous designs and methods. I do believe the authors' are being thoughtful in their approach, but also that the bar for epidemiologic rigor and ensuring relevant estimand needs to be quite high.

Comments from the reviewers:

Reviewer #1 (statistical review):

Thank you for addressing my comments. I have a few more things that need to be addressed:

In the responses to my comments, the authors stated that the analyses have now been updated to include 500 bootstrap samples. However, the manuscript still says 200 bootstrap samples. Was this updated? And please make sure all the changes outlined in the responses are followed through in the manuscript.

In the response to original comment 4, authors state "The immortal time period from days 0 to 28 is the same regardless of treatment assignment, so it does not lead to immortal time bias.". But this is not true. Even though there is the same period of time, there will be bias if the risk of the outcome or death in the 0-28 day period is different. At the very least, please can the authors describe the proportion of individuals with an outcome or who died between days 5 and 29 in each group. This will give us a better understanding of if there is bias.

In the censoring models, what is defined as a censoring event? Death should not be included, as it is not "censoring". The final two options are last visit to the EHR and 180 days after index. This censoring only needs to be adjusted for if the authors believe the is a systematic difference between the groups in the risk of censoring due to these reasons. I struggle to see this. It would be interesting if the authors could also do a sensitivity analysis where they include results without the IPC weights to understand how sensitive results are to this analytical strategy.

Reviewer #2:

Thank you for your response to my comments. I have no further suggestions.

Reviewer #4:

Thank you for addressing all my comments. I have no further questions.

---

---

## [Editor Report · Decision Letter 3]

17 Jul 2025

Dear Dr. Preiss,

Thank you very much for re-submitting your manuscript "Effect of Paxlovid Treatment during Acute COVID-19 on Long COVID Onset: An EHR-Based Target Trial Emulation from the N3C and RECOVER Consortia" (PMEDICINE-D-24-02349R3) for review by PLOS Medicine.

Thank you for your detailed response to the reviewers' and editors’ comments. I have discussed the paper with my colleagues, and it has also been seen again by the academic editor. The changes made to the paper were satisfactory to us and the academic editor. As such, we intend to accept the paper for publication, pending your attention to the reviewers' and editors' comments below in a further revision. When submitting your revised paper, please once again include a detailed point-by-point response to the editorial comments. The remaining issues that need to be addressed are listed at the end of this email.

In revising the manuscript for further consideration here, please ensure you address the specific points made by each reviewer and the editors. In your rebuttal letter you should indicate your response to the reviewers' and editors' comments and the changes you have made in the manuscript. Please submit a clean version of the paper as the main article file. A version with changes marked must also be uploaded as a marked up manuscript file. Please also check the guidelines for revised papers at http://journals.plos.org/plosmedicine/s/revising-your-manuscript for any that apply to your paper.

We ask that you submit your revision within 1 week (Jul 30 2025). However, if this deadline is not feasible, please contact me by email, and we can discuss a suitable alternative.

Please do not hesitate to contact me (atosun@plos.org) or us (plosmedicine@plos.org) directly with any questions. Please note that I will be out of the office from July 21 to August 1.

We look forward to receiving the revised manuscript.

Sincerely,

Alexandra Tosun, PhD

Senior Editor

PLOS Medicine

plosmedicine.org

Requests from Academic Editor:

The authors have done a thorough job addressing the prior comments and it is much appreciated. One small query. Is it possible to include vaccination status as a stratified analysis? This would help make the manuscript more comprehensive as this has come up in prior literature as an effect modifier. The intent is not necessarily that the analysis is discussed in much detail, but just having the stratified analysis as an exploratory look at potential effect modifiers.

I note that the authors briefly mentioned in the Methods section that ascertaining vaccination status in the EHR can be challenging. If they truly believe it cannot be done, it would be helpful for them to address this issue in the limitations section.

Requests from the Editors:

GENERAL

* Please confirm that your title complies with to PLOS Medicine's style. Your title must be nondeclarative and not a question. It should begin with main concept if possible. "Effect of" should be used only if causality can be inferred, i.e., for an RCT. Please place the study design ("A randomized controlled trial," "A retrospective study," "A modelling study," etc.) in the subtitle (i.e., after a colon).

* Statistical reporting: Please revise throughout the manuscript, including tables and figures.

- Please report statistical information as follows to improve clarity for the reader ""22% (95% CI [13,28]; p</=)"".

- Please separate upper and lower bounds with commas instead of hyphens as the latter can be confused with reporting of negative values.

- Please repeat statistical definitions (HR, CI etc.) for each set of parentheses.

* Please include the statement on code availability in the data availability statement.

* Please ensure that all abbreviations are defined at first use throughout the text (including statistical abbreviations). Please also check figures and tables.

* Please ensure that tables and figures, including those in supplementary files, are appropriately referenced in the main text.

* Please review your text for claims of novelty or primacy (e.g. 'for the first time' or ‘novel’) and remove this language.

* Please check that any use of statistical terms (such as trend or significant) are supported by the data, and if not please remove them. The term trend should be used only when the test for trend has been conducted.

ABSTRACT

* Please confirm that your abstract complies with our requirements, including providing all the information relevant to this study type https://journals.plos.org/plosmedicine/s/submission-guidelines#loc-abstract

* Abstract Background: The final sentence should clearly state the study question.

* Abstract Methods and Findings: We suggest clearly stating the primary and secondary outcomes.

* When reporting age, please include a unit (i.e., years).

* When describing the results, please mention that they are being compared to those of the control group.

* Please include the study setting, years during which the study took place, and baseline demographics.

* Please include the important dependent variables that are adjusted for in the analyses (or describe the overarching categories).

* Please ensure that all numbers presented in the abstract are present and identical to numbers presented in the main manuscript text.

INTRODUCTION

* Please conclude the Introduction with a clear description of the study question or hypothesis.

METHODS AND RESULTS

* Please ensure to cite the STROBE checklist. When completing the checklist, please use section and paragraph numbers, rather than text excerpts.

* Figure 1: We suggest defining U07.1 diagnosis in the figure description.

* Figures and Tables: Please ensure that all abbreviations are defined in figure/table descriptions.

* Are the results in Figure 3 repetitive of Table 3?

* “Among the age strata, Paxlovid had a significant effect on overall PASC onset only among patients aged 65+.” – what about the results for age group Age 18-24 and Age 35-49?

* What about the finding that there was an anti-protective effect in the sensitivity analysis for Positive Lab-only Index Events?

* Please include the Ethics Statement in the Methods section.

* Table 1: Please state in the Table that the results are ‘N (%)’.

* Table 3: For the columns ‘Relative Risk’ and ‘ Absolute Risk Difference’, please include ‘(95% CI)’ in the column headers.

General Editorial Requests

---

## [Editor Report · Decision Letter 4]

7 Aug 2025

Dear Dr Preiss, 

On behalf of my colleagues and the Academic Editor, Aaloke Mody, I am pleased to inform you that we have agreed to publish your manuscript "Effect of Paxlovid Treatment during Acute COVID-19 on Long COVID Onset: An EHR-Based Target Trial Emulation from the N3C and RECOVER Consortia" (PMEDICINE-D-24-02349R4) in PLOS Medicine.

I appreciate your thorough responses to the reviewers' and editors' comments throughout the editorial process. We look forward to publishing your manuscript, and editorially there are only a few remaining points that should be addressed prior to publication. We will carefully check whether the changes have been made. If you have any questions or concerns regarding these final requests, please feel free to contact me at atosun@plos.org.

Please see below the minor points that we request you respond to:

* Please ensure to update the online submission form with the finalized details, e.g. the updated data availability statement.

* “All results are reported in adherence with the Strengthening the Reporting of Observational Studies in Epidemiology (STROBE) guidelines” – please move this statement to the Methods section and ensure to cite the checklist in the main text.

* STROBE checklist: Please complete the checklist using section and paragraph numbers, rather than text excerpts.

* Please remove all subheadings from the Discussion section.

* Table 3 and Figure 3: Please separate upper and lower bounds with commas instead of hyphens as the latter can be confused with reporting of negative values.

Before your manuscript can be formally accepted you will need to complete some formatting changes, which you will receive in a follow up email (including the editorial requests above). Please be aware that it may take several days for you to receive this email; during this time no action is required by you. Once you have received these formatting requests, please note that your manuscript will not be scheduled for publication until you have made the required changes.

PRESS

Sincerely, 

Alexandra Tosun, PhD 

Senior Editor 

PLOS Medicine